# A FINE-GRAINED APPROACH TO EXPLAINING CATASTROPHIC FORGETTING OF INTERACTIONS IN CLASS-INCREMENTAL LEARNING

## ABSTRACT

This paper explains catastrophic forgetting in class incremental learning (CIL) from a novel perspective of interactions (non-linear relationship) between different input variables. Specifically, we make the first attempt to explicitly identify and quantify which interactions *w.r.t.* previous classes that are forgotten and preserved over incremental steps, and reveal their distinct behaviors, so as to provide a more fine-grained explanation of catastrophic forgetting. Based on the forgotten interactions, we provide a unified explanation for the effectiveness of some classical CIL methods in mitigating catastrophic forgetting, *i.e.,* these methods all reduce the forgetting of interactions *w.r.t.* previous classes, particularly those of low complexities, although these methods are originally designed based on different intuitions and observations. Intrigued by this, we further propose a simple-yet-efficient method with theoretical guarantees to investigate the role of low-complexity interactions in the resistance of catastrophic forgetting, and discover that low-order interaction serves as an effective factor in resisting catastrophic forgetting. *The code will be released if the paper is accepted.*

## 1 INTRODUCTION

Deep neural networks (DNNs) have been widely used in class incremental learning to solve a common real-world problem of learning new classes continually. However, they often suffer from catastrophic forgetting, *i.e.,* directly training the DNN to learn new classes will erase the knowledge of previous classes and result in a decline in performance. Thus, previous research mainly focused on proposing various CIL methods to mitigate catastrophic forgetting, such as regularization-based methods (Li & Hoiem, 2017; Zhao et al., 2020), memory-based methods (Zhou et al., 2023b; Lopez-Paz & Ranzato, 2017; Chaudhry et al., 2019), expansion-based methods (Yan et al., 2021; Wang et al., 2022; Zheng et al., 2025), and *etc*.

However, less attention has been devoted to explaining catastrophic forgetting, and existing studies typically employed accuracy-based metrics for explanation. For example, Chaudhry et al. (2018); Wang et al. (2024) computed the difference between the highest test accuracy on a previous task and the final test accuracy to measure the forgetting of this task. Lopez-Paz & Ranzato (2017) calculated the average accuracy drop of all previous tasks to evaluate the DNN's forgetting in the continual learning process. Whereas, these accuracy-based metrics can only provide a coarse-grained explanation for the **outcome** of the catastrophic forgetting, *i.e.,* the decline in the classification accuracy, but they fail to provide a fine-grained understanding of its underlying **cause**, *i.e.,* what knowledge is forgotten or preserved and to what extent during CIL.

Thus, unlike previous studies, this paper aims to provide a detailed explanation for catastrophic forgetting from a new perspective, by defining and extracting the knowledge encoded in an incrementally learned DNN, quantifying its forgetting of old knowledge over incremental steps, and summarizing its distinct property.

However, how to explicitly mathematically define the knowledge[1] encoded in the DNN still remains a challenge. To this end, **Ren et al. (2024a; 2023a) have derived a set of theorems as mathe-**

---

[1]Please see Appendix B for related works on extracting knowledge of the DNN.

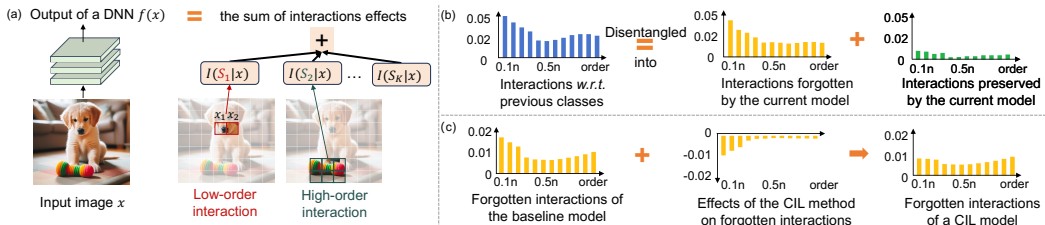

Figure 1: (a) The network output of a certain input sample is proven to be the sum of interaction effects. Each interaction refers to an AND relationship between a set of input variables. (b) We identify and quantify interactions of previous classes that are forgotten and preserved over incremental steps to detailedly explain catastrophic forgetting. (c) We use forgotten interactions to explain the shared mechanism of different CIL methods in mitigate catastrophic forgetting, *i.e.,* reducing the forgetting of interactions of previous classes, particularly low-order interactions.

**matical evidence to faithfully take countable interactions between different input variables to represent uncountable knowledge encoded by the DNN.** Specifically, given a sample $x$, a DNN usually does not employ each single input variable of $x$ independently for prediction. Instead, the DNN lets each input variable interact with each other to form a certain pattern for inference. For a better understanding, let us consider the toy example in Fig. 1. The DNN encodes the interaction between two image patches in $S = \{x_1, x_2\}$ to form a *dog nose* pattern. Each interaction represents an AND relationship between variables in $S$. That is, only when all two variables in $S$ are all present, the *dog nose* interaction is activated, and make a numerical effect $I(S)$ on the network output. The masking[2] of any patch in $S$ will deactivate this *dog nose* interaction, and removes its numerical effect $I(S)$.

More crucially, let us randomly mask this input sample $x$ in different ways to generate different masked samples $\{x_{\text{masked}}\}$, by randomly masking some input variables and keeping other variables unchanged. Ren et al. (2024a; 2023a) have proven that people can use a few interactions to accurately approximate the DNN's outputs on all these masked samples $f(x_{\text{masked}})$, as shown in Fig. 1, which serves as mathematical evidence to justify the convincingness of considering above interactions to represent the knowledge encoded by the DNN for inference.

Surprisingly, we find interactions can explain different aspects of catastrophic forgetting, which may help both practitioners and theoreticians gain some new insights into the learning dynamics of an incrementally trained DNN. Specifically, this paper aims to answer the following three questions.

**(1) How to identify what is forgotten in class incremental learning.** To address this, given a current step $t$ and a previous step $k \in \{1, 2, \cdots, t-1\}$, we propose metrics to disentangle the interaction of each complexity encoded by the previous DNN $f_k$ (incrementally trained from steps 1 to $k$) into interaction components forgotten and preserved by the current DNN $f_t$. As shown in Fig. 1 (b), a large number of interactions *w.r.t.* previous classes are forgotten, particularly those of low complexity, while only a small number are preserved, ultimately resulting in catastrophic forgetting. This quantitative finding provide more direct and concrete evidence to support the widely accepted qualitative claim that the erasure of a large amount of knowledge *w.r.t.* previous classes leads to catastrophic forgetting (Zhou et al., 2024a), compared to previous accuracy-based metrics. Notably, the complexity (or *order*) of an interaction is defined as the number of variables involved in this interaction $S$. As Fig. 1 shows, a low-order interaction usually represents the simple AND relationship between a few variables.

**(2) How to explain the effectiveness of different CIL methods in mitigating catastrophic forgetting in a unified view.** Based on the above disentangled forgotten interactions, we compare and discover a clear difference between the DNN trained with a certain CIL method to mitigate catastrophic forgetting (abbreviated as the *CIL model*) and the DNN trained solely with the cross-entropy loss, without any anti-forgetting method (abbreviated as the *baseline model*). As shown in Fig. 1 (c), Fig. 3, and Fig. 4, different CIL methods all share a common mechanism that they all make the the CIL model reduce the forgetting of interactions *w.r.t.* previous classes, particularly the forgetting of low-order interactions, to resist catastrophic forgetting, although they are originally designed based on different observations and intuitions. Thus, our paper makes the **first** attempt to **unify** the effectiveness of different anti-forgetting CIL methods in a single theoretic system.

**(3) What role low-order interactions play in the resistance of catastrophic forgetting.** To investigate this, we propose a simple-yet-efficient method to penalize an incrementally learned DNN from encoding low-orders, and compare its stability to that of a DNN encoding low-order interactions, where the stability refers to the DNN's ability to resist catastrophic forgetting (Wang et al., 2024). In experiments, we discover that low-order interactions may, to some extent, serve as an effective factor for resisting catastrophic forgetting.

**Contributions** of this paper are summarized as follows. (1) We make the first attempt to disentangle and quantify the forgotten interactions of different complexities in the incremental learning process, so as to provide a detailed explanation for catastrophic forgetting. (2) We reveal the unified mechanism of different CIL methods in mitigating catastrophic forgetting. (3) We explain the role of low-order interactions in resisting catastrophic forgetting.

## 2   PRELIMINARIES: USING INTERACTIONS TO REPRESENT KNOWLEDGE IN DNNs

Although there is no consensus on the definition of knowledge encoded by a DNN so far, Ren et al. (2024a; 2023a) have proven a set of properties to mathematically justify why we can take interactions to represent knowledge in a DNN.

**Definition of interactions.** Given a trained DNN $f$, let $\boldsymbol{x} \in \mathbb{R}^n$ denote an input sample consisting of $n$ input variables totally, and $N = \{1, 2, \cdots, n\}$ denote the indices of all variables. The input variable can be defined differently depending on the task, *e.g.,* it can represent an image patches for image classification or a word/token for text classification. Let $f(\boldsymbol{x}) \in \mathbb{R}$ represent the scalar network output or a certain output dimension of the DNN. Note that people can apply different settings for $f(\boldsymbol{x})$. Here, we follow (Ren et al., 2024a; Li & Zhang, 2023; Ren et al., 2023a) to set $f(\boldsymbol{x})$ as the confidence of predicting $\boldsymbol{x}$ to the ground-truth category $y^{\text{truth}}$ in multi-category classification tasks.

$$f(\boldsymbol{x}) = \log \left( p(y = y^{\text{truth}}|\boldsymbol{x})/(1 - p(y = y^{\text{truth}}|\boldsymbol{x})) \right), \tag{1}$$

Then, Ren et al. (2023a) employed the Harsanyi dividend (Harsanyi, 1963), a typical metric in game theory, to quantify the numerical effect of the interaction between variables inside a subset $S \subseteq N$ on the network output $f$.

$$I(S|\boldsymbol{x}) = \sum\nolimits_{U \subseteq S} (-1)^{|S|-|U|} \cdot f(\boldsymbol{x}_U), \tag{2}$$

where $\boldsymbol{x}_U$ denotes a masked sample crafted by masking[2] input variables in $N \setminus U$ and keeping variables in $U$ unchanged. Thus, $f(\boldsymbol{x}_U)$ represents the network output on the masked sample $\boldsymbol{x}_U$.

**Understanding of interactions.** Each interaction refers to an AND relationship between variables in the subset $S$. For instance, let us consider the interaction inside the subset $S = \{x_1, x_2\}$ shown in Fig. 1, which forms a semantic pattern of *dog nose*. Only when all two input variables in $S$ are all present, the nose interaction is triggered, and contribute a numerical effect $I(S|\boldsymbol{x})$ on the network output. Otherwise, the masking of any variable will break the nose interaction, and remove the numerical effect $I(S|\boldsymbol{x})$, *i.e.,* $I(S|\boldsymbol{x}) = 0$.

**Faithfulness of the interaction-based explanation.** The following four properties serve as convincing evidence that interactions can faithfully represent knowledge encoded by a DNN, rather than a mathematical formulation without clear meanings, since the inference logic of the DNN can be well explained by interactions.

**(1) Sparsity property.** Given a sample $\boldsymbol{x} \in \mathbb{R}^n$, a DNN theoretically can encode at most $2^n$ different interactions[3] *w.r.t.* $2^n$ different subsets $S \subseteq N$. However, Ren et al. (2024a) have proven that a well-trained DNN for classification usually encode very sparse salient interactions, which has also been widely observed on various DNNs for different tasks (Ren et al., 2023a;b; Zhou et al., 2024b; Cheng et al., 2024; Ren et al., 2025). We also observe this on DNNs trained for CIL (cf. Appendix G).

---

[2]In practice of masking input variables in $N \setminus U$, people commonly use baseline values $\{b_i\}$ to replace their original values (Ancona et al., 2019; Covert et al., 2020; Ren et al., 2024a), *i.e.,* setting $x_i = b_i$ if $i \in N \setminus U$.

[3]We follow the method in (Li & Zhang, 2023; Ren et al., 2023a) to accelerate the computation of interactions of each input sample, which takes only a few seconds with a single NVIDIA 4090 GPU. Please see Appendix H for computation details and specific time costs.

**Theorem 1** (Sparsity property, proven in (Ren et al., 2024a)). *Given an input sample $\boldsymbol{x} \in \mathbb{R}^n$, let $\Omega_{salient} = \{S | S \subseteq N, |I(S|\boldsymbol{x})| \geq \tau\}$ denote a set of salient interactions whose absolute value exceeds a threshold $\tau$. If the DNN can generate relatively smooth inference outputs $f(\boldsymbol{x}_T)$ on masked samples[4], then the size of the set $|\Omega_{salient}|$ is proven to have an upper bound of $\mathcal{O}(n^{\kappa}/\tau)$, where $\kappa$ is an intrinsic parameter for the smoothness of the network function $f(\cdot)$. Empirically, $\kappa$ is usually within the range of $[1.9, 2.2]$ (Ren et al., 2024b), thus $|\Omega_{salient}|$ is much less than $2^n$.*

**(2) Universal matching property.** Theorem 2 proves that we can use a few salient interactions in $\Omega_{\text{salient}}$ to universally match network outputs $f(\boldsymbol{x}_S)$ on all $2^n$ masked samples.

**Theorem 2** (Universal matching property, proven in (Ren et al., 2023a)). *The network output $f(\boldsymbol{x}_S)$ on each masked sample $\{\boldsymbol{x}_S | \forall S \subseteq N\}$ is proven to be well mimicked by the sum of interaction effects.*

$$f(\boldsymbol{x}_S) = \sum\nolimits_{U \subseteq S} I(U|\boldsymbol{x}) \approx \sum\nolimits_{U \subseteq S \& U \in \Omega_{salient}} I(U|\boldsymbol{x}). \tag{3}$$

**(3) Transferability property.** Li & Zhang (2023) has observed that DNNs with different architectures learned under the same training strategy for the same task usually encode similar sets of interactions for inference.

**(4) Discrimination property.** Li & Zhang (2023) has discovered that interactions usually exhibit considerable discrimination power, *i.e.,* the same interactions extracted from different samples usually pushes the DNN towards the classification of the same category.

**Complexity of the interaction.** The complexity (or *order*) of an interaction is defined as the number of variables involved in this interaction $S$ (Ren et al., 2023a; 2024a), *i.e.,* order $= |S|$. As Fig. 1 shows, a low-order interaction represents simple AND relationship between a few variables, while a high-order interaction refers to complex AND relationship between numerous variables.

# 3 EXPLAINING CATASTROPHIC FORGETTING USING INTERACTIONS

Let us first revisit the class incremental learning, which aims to train a DNN on data that arrives incrementally with new classes (Rebuffi et al., 2017). Specifically, given a sequence of $T$ steps $\mathcal{D} = \{\mathcal{D}_1, \mathcal{D}_2, \cdots, \mathcal{D}_T\}$ without overlapping classes, the $t$-th step $\mathcal{D}_t = \{(\boldsymbol{x}_{j,t}, y_{j,t})\}_{j=1}^{N_t}$ contains $N_t$ tuples of input samples $\boldsymbol{x}_{j,t} \in \mathbb{R}^n$ and its corresponding label $y_{j,t}$. Then, class incremental learning is to optimize the DNN to minimize the average loss of all classes, where the data from previous training steps is not available or restricted during the current training step (Zhou et al., 2024a).

Owing to the guaranteed faithfulness of the interaction, we use interactions to (i) quantify what knowledge is forgotten and preserved throughout the CIL process in Section 3.1, (ii) provide a unified understanding of the effectiveness of different CIL methods in Section 3.2, and (iii) reveal the crucial role of low-order interactions in mitigating catastrophic forgetting in Section 3.3.

## 3.1 QUANTIFYING FORGOTTEN INTERACTIONS IN CIL

In this section, we make the first attempt to quantitatively identify which interactions used for the inference of previous classes are forgotten over incremental steps and ultimately lead to catastrophic forgetting, while accuracy-based forgetting metrics (Chaudhry et al., 2018; Wang et al., 2024) fail to reveal what is forgotten and to what extent, limiting deeper insights into catastrophic forgetting.

To this end, given a current step $t$ and a previous step $k \in \{1, \cdots, t-1\}$, we consider that the DNN $f_k$, trained incrementally on $\{\mathcal{D}_i\}_{i=1}^k$, encodes richest interactions relevant to the inference of $\mathcal{D}_k$. However, these interactions are gradually forgotten over incremental steps, with some preserved and utilized by subsequent DNNs $\{f_t\}$ for classification. Explicitly speaking, let us use $\mathcal{I}^m(S|\boldsymbol{x}_k, f_k) \triangleq |I^m(S|\boldsymbol{x}_k, f_k)|$ to quantify the strength of the $m$-order interaction $I^m(S|\boldsymbol{x}_k, f_k)$ of the sample $\boldsymbol{x}_k \in \mathcal{D}_k$ encoded by the DNN $f_k$. We propose the following two metrics to disentangle $\mathcal{I}^m(S|\boldsymbol{x}_k, f_k)$ into interaction components preserved $\mathcal{I}_{\text{preserve}}^{(k,t),m}(S|\boldsymbol{x}_k)$ and forgotten $\mathcal{I}_{\text{forget}}^{(k,t),m}(S|\boldsymbol{x}_k)$ by the current DNN $f_t$, where $\mathcal{I}_{\text{preserve}}^{(k,t),m}(S|\boldsymbol{x}_k)$ quantifies the preserved interaction shared by both $f_t$ and $f_k$, and

---

[4]This is formulated by three common mathematical conditions. Please see Appendix G for details.

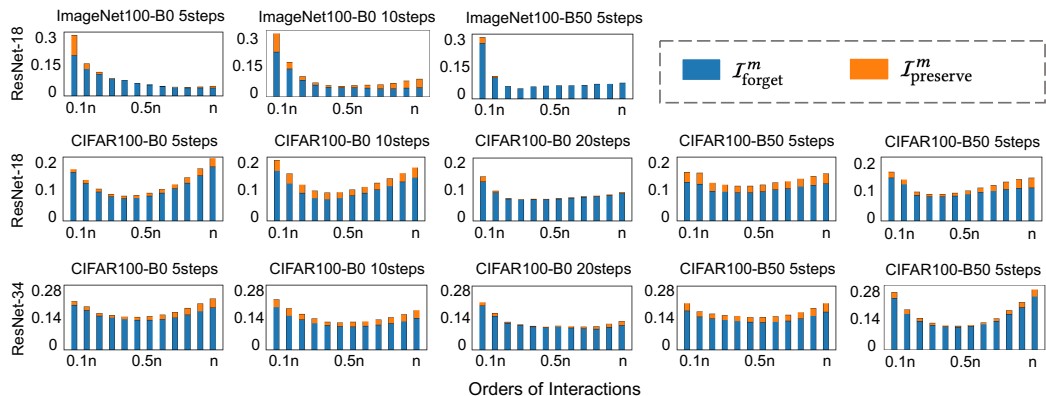

Figure 2: The forgotten ($\mathcal{I}_{\text{forget}}^m$) and preserved ($\mathcal{I}_{\text{preserve}}^m$) interactions *w.r.t.* previous classes.

$\mathcal{I}_{\text{forget}}^{(k,t),m}(S|\boldsymbol{x}_k)$ measures the interaction encoded in $f_k$ but later forgotten by $f_t$.

$$\mathcal{I}_{\text{preserve}}^{(k,t),m}(S|\boldsymbol{x}_k) = \Gamma_k^t(S|\boldsymbol{x}_k) \cdot \min(\mathcal{I}^m(S|\boldsymbol{x}_k, f_k), \mathcal{I}^m(S|\boldsymbol{x}_k, f_t)),$$
$$\mathcal{I}_{\text{forget}}^{(k,t),m}(S|\boldsymbol{x}_k) = \mathcal{I}^m(S|\boldsymbol{x}_k, f_k) - \mathcal{I}_{\text{preserve}}^{(k,t),m}(S|\boldsymbol{x}_k),$$

(4)

where $\mathcal{I}^m(S|\boldsymbol{x}_k, f_k)$ and $\mathcal{I}^m(S|\boldsymbol{x}_k, f_t)$ denote the strength of $m$-order interaction extracted from the input sample $\boldsymbol{x}_k \in \mathcal{D}_k$ encoded by the DNNs $f_k$ and $f_t$, respectively, which is calculated based on Eq. (1) and Eq. (2), and $|S| = m$. $\Gamma_k^t(S|\boldsymbol{x}_k) = \mathbb{1}((I^m(S|\boldsymbol{x}_k, f_k) \cdot I^m(S|\boldsymbol{x}_k, f_t)) > 0)$ measures whether the interaction $I^m(S|\boldsymbol{x}_k, f_t)$ has the same effect as the interaction $I^m(S|\boldsymbol{x}_k, f_k)$, where $\mathbb{1}(\cdot)$ is the indicator function. If $I^m(S|\boldsymbol{x}_k, f_t)$ and $I^m(S|\boldsymbol{x}_k, f_k)$ have opposite effects, then the preserved interaction $\mathcal{I}_{\text{preserve}}^{(k,t),m}(S|\boldsymbol{x}_k) = 0$, indicating the current DNN $f_t$ completely forgets the $m$-order interaction learned at the step $k$. Otherwise, the preserved interaction is quantified as the shared component $\mathcal{I}_{\text{preserve}}^{(k,t),m}(S|\boldsymbol{x}_k) = \min(\mathcal{I}^m(S|\boldsymbol{x}_k, f_k), \mathcal{I}^m(S|\boldsymbol{x}_k, f_t))$.

Hence, based on Eq. (4), we can identify which interactions of each complexity *w.r.t.* previous steps $k$ are forgotten or preserved during the entire $T$-step class-incremental learning process, as well as quantifying their exact amounts, as follows.

$$\mathcal{I}_{\text{forget}}^m = \frac{1}{T-2} \sum_{t=2}^T \left( \frac{1}{t-1} \sum_{k=1}^{t-1} \mathbb{E}_{\boldsymbol{x}_k \in \mathcal{D}_k} \mathbb{E}_{S \subseteq N, |S|=m} \left[ \mathcal{I}_{\text{forget}}^{(k,t),m}(S|\boldsymbol{x}_k) \right] \right),$$
$$\mathcal{I}_{\text{preserve}}^m = \frac{1}{T-2} \sum_{t=2}^T \left( \frac{1}{t-1} \sum_{k=1}^{t-1} \mathbb{E}_{\boldsymbol{x}_k \in \mathcal{D}_k} \mathbb{E}_{S \subseteq N, |S|=m} \left[ \mathcal{I}_{\text{preserve}}^{(k,t),m}(S|\boldsymbol{x}_k) \right] \right).$$

(5)

**Experiments.** We conducted experiments to explain how an incrementally trained DNNs forgot and preserved interactions *w.r.t.* previous classes. Notably, we focused on the conventional CIL setting, where catastrophic forgetting was typically more pronounced. To this end, we followed settings in (Rebuffi et al., 2017; Hou et al., 2019) to incrementally train ResNet-18 and ResNet-34 models (He et al., 2016) on the CIFAR-100 dataset (Krizhevsky et al., 2009), and ResNet-18 model on the ImageNet-100 dataset (Rebuffi et al., 2017) under different class splits. Specifically, we employed two widely-used class splits to train DNNs. First, we followed (Rebuffi et al., 2017) to equally divide all classes of the CIFAR-100 dataset into 5, 10, and 20 incremental steps (denoted as ***CIFAR100-B0***), and all classes of the ImageNet-100 dataset into 5 and 10 incrementally steps (denoted as ***ImageNet100-B0***). Second, we followed (Hou et al., 2019) to allocate half of the total classes to the first step, and further equally divide the rest classes of the CIFAR-100 dataset into 5 and 10 incrementally steps, and divide the remaining classes of the ImageNet-100 dataset into 5 incrementally steps. These settings were referred to as ***CIFAR100-B50*** and ***ImageNet100-B50***, respectively. Please see Appendix H for more experimental details.

Fig. 2 reports the forgotten $\mathcal{I}_{\text{forget}}^m$ (blue bars) and preserved $\mathcal{I}_{\text{preserved}}^m$ (orange bars) interactions of different orders $m$. We discovered that for each incrementally learned DNN, $\mathcal{I}_{\text{forget}}^m$ consistently exceeded $\mathcal{I}_{\text{preserve}}^m$. Explicitly speaking, the strength of forgotten interactions was on average approximately 6 times that of preserved interactions, indicating that the DNN suffered severe forgetting of a large number of interactions *w.r.t.* previous classes in CIL. This phenomenon echoed and provided

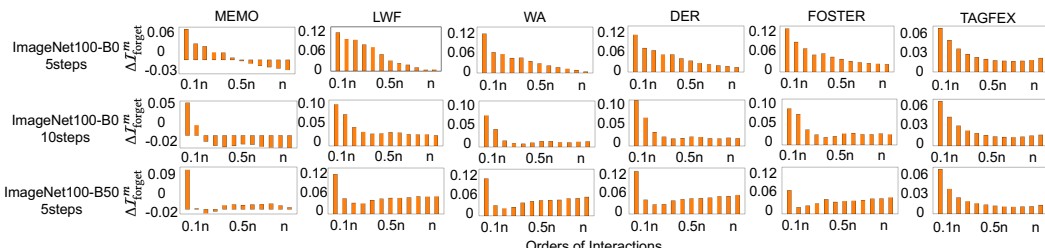

Figure 3: The difference in forgotten interactions $\Delta\mathcal{I}_{\text{forget}}^m$ *w.r.t.* previous classes between the CIL model and the baseline model, both incrementally trained on the ImageNet-100 dataset.

concrete quantitative support for the widely accepted qualitative claim that the erasure of a large amount of knowledge *w.r.t.* previous classes resulted in catastrophic forgetting (Zhou et al., 2024a).

Besides, we also observed from Fig. 2 that the incrementally learned DNN was prone to forgetting relatively more low-order interactions *w.r.t.* previous classes, which indicated that these low-order interactions were relatively more easily forgotten in class incremental learning. In comparison, we found that different DNNs exhibited relatively different tendencies in preserving interactions.

## 3.2 A UNIFIED EXPLANATION FOR CIL METHODS

In this section, we use forgotten interactions to provide a **unified** explanation for the effectiveness of different CIL methods in mitigating catastrophic forgetting, *i.e.,* whether and how differently designed CIL methods help the DNN resist the forgetting of such interactions.

To this end, let us consider two DNNs $f_{\text{base}}$ and $f_{\text{CIL}}$, sharing the same architecture and both incrementally trained on $\{\mathcal{D}_i\}_{i=1}^T$. Among them, $f_{\text{CIL}}$ is trained with a certain CIL method (*e.g.,* TagFex (Zheng et al., 2025)) to resist catastrophic forgetting, which is abbreviated as the ***CIL model***. In comparison, $f_{\text{base}}$ is trained merely with the cross-entropy loss, without any anti-forgetting methods, which is abbreviated as the ***baseline model***. Then, we compare the difference between $m$-order interactions *w.r.t.* previous classes forgotten by the CIL model $\mathcal{I}_{\text{forget,CIL}}$ with those forgotten by the baseline model $\mathcal{I}_{\text{forget,base}}$, to explain the effectiveness of the CIL model in mitigating catastrophic forgetting.

$$\Delta\mathcal{I}_{\text{forget}}^m = \mathcal{I}_{\text{forget,base}}^m - \mathcal{I}_{\text{forget, CIL}}^m, \tag{6}$$

where $\mathcal{I}_{\text{forget,CIL}}$ and $\mathcal{I}_{\text{forget,base}}$ are computed based on Eq. (5) with $f_k = f_{k,\text{base}}, f_t = f_{t,\text{base}}$ for the baseline model and $f_k = f_{k,\text{CIL}}, f_t = f_{t,\text{CIL}}$ for the CIL model, respectively. A positive value of $\Delta\mathcal{I}_{\text{forget}}^m$ indicates the CIL model forgets fewer $m$-order interactions *w.r.t* previous classes than the baseline model, and vice versa.

**Experiments.** To explain the effectiveness of CIL methods, we trained ResNet-18 and ResNet-34 models on CIFAR-100 dataset, and learned ResNet-18 model on ImageNet-100 dataset under different class spilts in Section 3.1 for class incremental learning. Considering existing CIL methods could be divided into different categories (Liang & Li, 2023), we picked several classical and open-sourced CIL methods for each category to ensure the generality of our explanation, *i.e.,* LWF (Li & Hoiem, 2017) and WA (Zhao et al., 2020) for regularization-based, MEMO (Zhou et al., 2023b) for memory-based, DER (Yan et al., 2021), FOSTER (Wang et al., 2022), and TagFex (Zheng et al., 2025) for expansion-based, and DS-AL (Zhuang et al., 2024) for analytic-learning-based. Thus, for each DNN, we trained 8 versions based on PyCIL (Zhou et al., 2023a), including a baseline model learned merely with cross-entropy loss, and **7** CIL models trained with different CIL methods.

Fig. 3 and Fig. 4 reports the difference in forgotten interactions $\Delta\mathcal{I}_{\text{forget}}^m$ between the CIL model and the baseline model. We discovered that the CIL model mitigated the forgetting of interactions $\mathcal{I}_{\text{forget, CIL}}^m$ *w.r.t.* previous classes compared to the baseline model, *i.e.,* $\mathbb{E}_m[\Delta\mathcal{I}_{\text{forget}}^m] > 0$. Interestingly, such a phenomenon was shared by different CIL models, although they were not originally designed for this purpose, but based on different intuitions and observations. Moreover, this shared phenomenon was also observed in a different CIL scenario, *i.e.,* audio-visual CIL with autoencoders (Pian et al., 2023), supporting its generality (cf. Appendix D for details). Thus, this shared phenomenon provided a unified understanding for the effectiveness of different CIL methods, *i.e.,* all making the incrementally learned DNN forget less interactions *w.r.t.* previous classes to mitigate

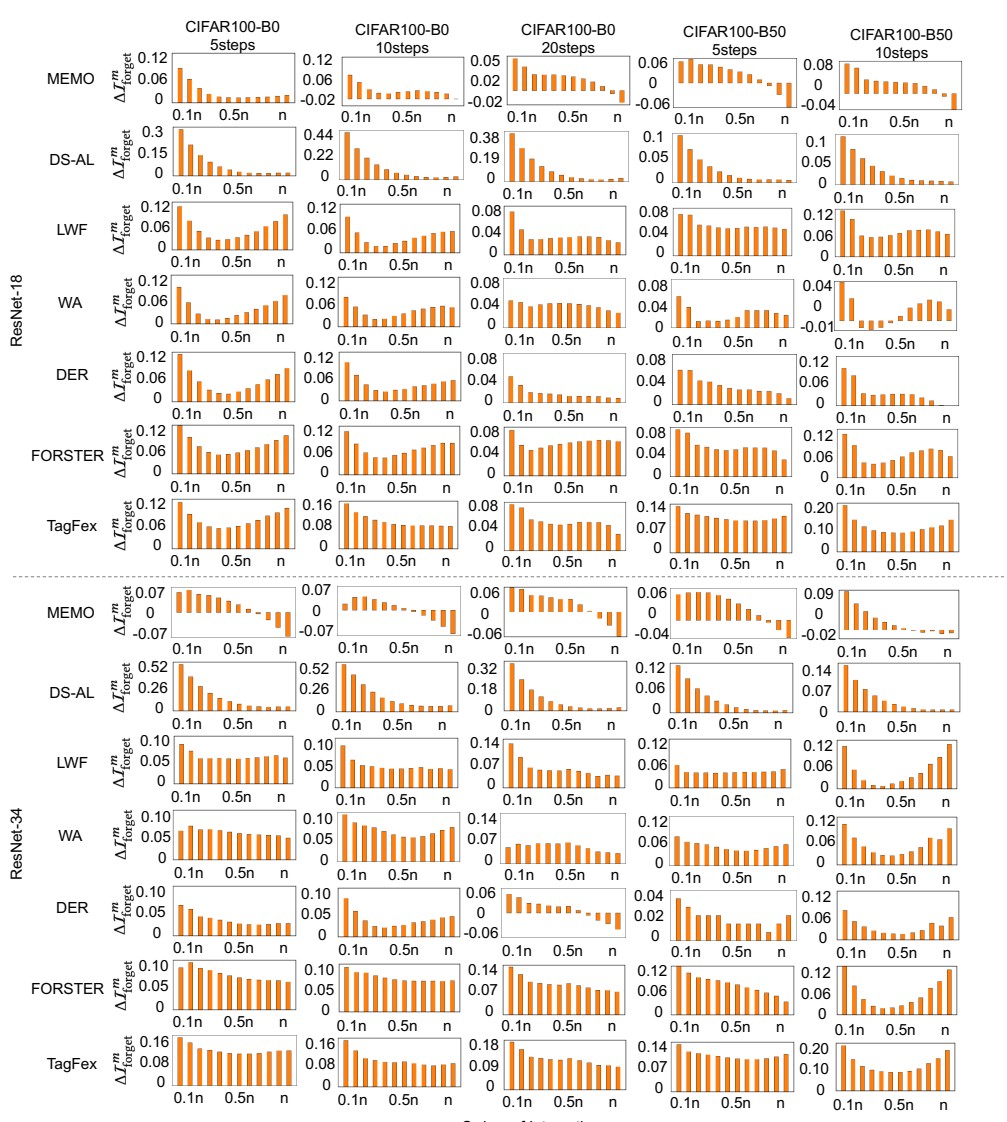

Figure 4: The difference in forgotten interactions $\Delta\mathcal{I}^m_{\text{forget}}$ *w.r.t.* previous classes between the CIL model and the baseline model, both incrementally trained on the CIFAR-100 dataset.

catastrophic forgetting, which also partially echoed the conclusion in Section 3.1 that the forgetting of old interactions $\mathcal{I}^m_{\text{forget}}$ could result in catastrophic forgetting.

Besides, we also observed another phenomenon that these CIL methods all made the DNN significantly mitigate the forgetting of low-order interactions, *i.e.*, low-order $\Delta\mathcal{I}^m_{\text{forget}}$ was consistently positive, and generally larger than that of higher orders. This indicated CIL models usually forgot significantly fewer low-order interactions than the baseline model, which could be intuitively understood as follows. Low-order interactions usually represented local and common features with considerable generalization power (Zhou et al., 2024b), and high-order interactions (usually global features) were usually constructed by low-order interactions. Thus, low-order interactions learned in previous steps could not only contribute to the inference of previous data, but also generalize and combine with low-order interactions newly learned in the current step to form high-order interactions for current data inference.

Notably, the CIL model trained with MEMO often forgot more high-order interactions *w.r.t.* previous classes than the baseline model, *i.e.*, $\Delta\mathcal{I}^m_{\text{forget}} < 0$ for high order $m$. This might be because MEMO made the DNN learn more new high-order interactions *w.r.t.* current classes, as shown in Fig. 7 of Appendix E, where we proposed a new metric to quantify the learning of new interactions at each step. Please refer to Appendix E for detailed discussion and results.

### 3.3 Exploring the Role of Low-Order Interactions in Resisting Catastrophic Forgetting

Intrigued by the above shared phenomenon that different CIL methods all forget significantly fewer low-order interactions, in this section, we further explore the role of low-order interactions in CIL. To this end, we propose a simple-yet-efficient method to investigate whether low-order interactions have a potential influence on the resistance of catastrophic forgetting, *i.e.,* checking whether the stability of an incrementally trained DNN significantly decreases when we prevent it from encoding low-order interactions, where the stability measures its ability to resist catastrophic forgetting (Wang et al., 2024). This provides new insights into CIL.

Specifically, we design a loss function to force the DNN to encode interactions of specific orders. According to the *universal matching property* in Theorem 2, the network output $f$ can be decomposed into the sum of interaction effects of varying orders. Thus, let us first focus on the network output change $\Delta f(m_1, m_2)$ between different masked samples $\{\boldsymbol{x}_{S_1}\}$ and $\{\boldsymbol{x}_{S_2}\}$, which serves as the foundation for designing the loss function.

$$\Delta f(m_1, m_2) = \mathbb{E}_{S_1, S_2}[f(\boldsymbol{x}_{S_2}) - f(\boldsymbol{x}_{S_1})], \quad \emptyset \subseteq S_1 \subsetneq S_2 \subseteq N, \ |S_1| = m_1 n, \ |S_2| = m_2 n, \quad (7)$$

where subsets $S_1$ and $S_2$ are randomly sampled from all input variables $N$, containing $m_1 n$ and $m_2 n$ variables, respectively, such that $\emptyset \subseteq S_1 \subsetneq S_2 \subseteq N$, and $0 \le m_1 \le m_2 \le 1$. Then, we have proven that the output change $\Delta f(m_1, m_2)$ mainly encodes interactions of $[0, m_2 n]$ orders, as follows.

**Theorem 3** (proven in Appendix C). *The change of the network output $\Delta f(m_1, m_2)$ is proven to be decomposed into interaction effects of different orders.*

$$\Delta f(m_1, m_2) = \sum_{m=0}^{n} w^{(m)} \cdot \mathbb{E}_{S \subseteq N, |S|=m}[I(S|\boldsymbol{x})],$$

$$w^{(m)} = \begin{cases} C_{m_2 n}^m - C_{m_1 n}^m, & m \le m_1 n, \\ C_{m_2 n}^m, & m_1 n < m \le m_2 n, \\ 0, & m_2 n < m \le n. \end{cases} \quad (8)$$

In this way, based on Theorem 3, we propose the following loss function to prevent the DNN from using $[0, m_2 n]$-order interactions encoded in $\Delta f(m_1, m_2)$ for inference, thereby penalizing the encoding of these interactions. Explicitly speaking, we maximized the classification cross entropy $L_{\text{inter}}(m_1, m_2)$ based on $\Delta f(m_1, m_2)$, in order to make $\Delta f(m_1, m_2)$ non-discriminative.

$$L_{\text{inter}}(m_1, m_2) = -\mathbb{E}_{\boldsymbol{x}} \left[ \sum_{c=1}^{C} \left[ p(\hat{y} = c|\Delta f_c(m_1, m_2, \boldsymbol{x})) \cdot \log(p(\hat{y} = c|\Delta f_c(m_1, m_2, \boldsymbol{x}))) \right] \right], \quad (9)$$

where $\Delta f_c(m_1, m_2, \boldsymbol{x}) = f_c(\boldsymbol{x}_{S_2}) - f_c(\boldsymbol{x}_{S_1})$ represents the change of the logits *w.r.t.* the category $c$. $C$ and $\hat{y}$ are referred to as the total number of classes and the predicted label, respectively. We compute the probability $p(\hat{y} = c|\Delta f_c(m_1, m_2, \boldsymbol{x}))$ of classifying the input sample $\boldsymbol{x}$ to a certain category $c$ by feeding the vector $[\Delta f_1(m_1, m_2, \boldsymbol{x}), \Delta f_2(m_1, m_2, \boldsymbol{x}), \cdots, \Delta f_C(m_1, m_2, \boldsymbol{x})]$ into the softmax layer.

Thus, we minimize the following loss function $L$ to train a DNN, forcing it to use interaction of specific orders for classification, where $L_{\text{classification}}$ is employed as the cross-entropy loss in practice, and the positive constant $\alpha$ is used to balance two loss terms.

$$L(m_1, m_2) = L_{\text{classification}} - \alpha \cdot L_{\text{inter}}(m_1, m_2). \quad (10)$$

**Experiment 1: investigating effects of the loss $L_{\text{inter}}(m_1, m_2)$.** Before explaining the role of $m$-order interactions in mitigating catastrophic forgetting, we first examine whether $L(m_1, m_2)$ could prevent the incrementally learned DNN from encoding interactions of specific orders. To this end, we trained ResNet-18 and ResNet-34 models on CIFAR-100 dataset under different class splits in Section 3.1 for CIL. For each DNN, we trained three versions, including a normally trained DNN without any interaction penalization by setting $\alpha = 0$ (*i.e.,* the baseline model in Section 3.2), and two DNNs trained with $\alpha = 1.0$ to penalize interactions of specific orders, setting $m_1 = 0, m_2 = 0.3$ and $m_1 = 0.7, m_2 = 1.0$ in $L(m_1, m_2)$, respectively. This training process is summarized in Algorithm 1 of Appendix H.

Fig. 5 reports the average interaction strength $\mathcal{I}^m$ of each DNN, $\mathcal{I}^m = \mathbb{E}_{\boldsymbol{x}} \mathbb{E}_{S \subseteq N, |S|=m}[|I^m(S|\boldsymbol{x})|]$. We discovered that $L_{\text{inter}}(m_1, m_2)$ could successfully prevent DNNs from encoding $[m_1 n, m_2 n]$-order interactions, instead of $[0, m_2 n]$-orders. That is, when disencouraging the DNN to encode interactions $[0, 0.3n]$ orders (or $[0.7n, n]$ orders), the interaction strength $I^{(m)}$ of $[0, 0.3n]$ orders (or $[0.7n, n]$

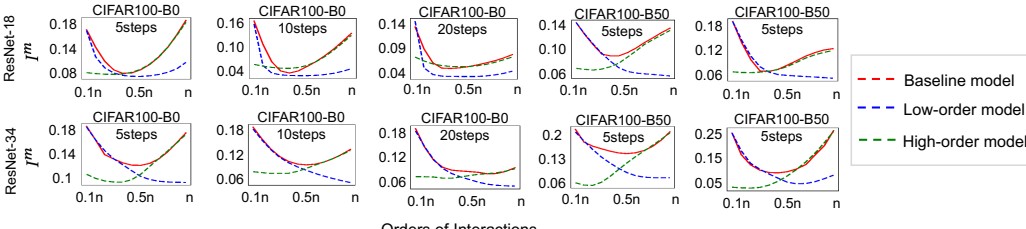

Figure 5: The distribution of interaction strength $\mathcal{I}^m$ of the baseline model, the low-order model, and the high-order model.

Table 1: The stability difference $\Delta FM_{\text{low}}$ between the low-order model and the baseline model, as well as $\Delta FM_{\text{high}}$ between the high-order model and the baseline model.

| Model | Metric | CIFAR100-B0 5 steps | CIFAR100-B0 10 steps | CIFAR100-B0 20 steps | CIFAR100-B50 5 steps | CIFAR100-B50 10 steps |
|---|---|---|---|---|---|---|
| ResNet-18 | $\Delta FM_{\text{low}}$ | 0.08 | 0.08 | 0.06 | 0.02 | 0.16 |
| | $\Delta FM_{\text{high}}$ | 0.22 | 0.21 | 0.24 | 0.23 | 0.31 |
| ResNet-34 | $\Delta FM_{\text{low}}$ | 0.10 | 0.05 | 0.08 | 0.14 | 0.08 |
| | $\Delta FM_{\text{high}}$ | 0.22 | 0.30 | 0.28 | 0.30 | 0.34 |

orders) were significantly decreased, compared to the baseline model. Thus, we named the DNN trained to penalize $[0, 0.3n]$-order interactions **high-order model** for simplicity, as it mainly encoded high-order interactions. Accordingly, the DNN trained to penalize $[0.7n, n]$-order interactions was termed the **low-order model**.

**Experiment 2: exploring the role of low-order interactions in mitigating catastrophic forgetting.** To this end, we compared the stability of the baseline model with that of DNNs trained to penalize interactions of specific orders. Specifically, we used the Forgetting Measure metric $FM$ (Chaudhry et al., 2018; Wang et al., 2024) to evaluate the stability of an incrementally learned DNN, which quantified the average decline in the performance on each previous step $k$.

$$FM = \mathbb{E}_{k=1}^{T-1}[\max_{i \in \{1, \cdots, T-1\}}(Acc^{(k,i)} - Acc^{(k,T)})], \tag{11}$$

where $Acc^{(k,i)}$ denoted the classification accuracy evaluated on the data $\mathcal{D}_k$ by the DNN $f_i$ incrementally trained over $i$ steps. A large $FM$ value implied low stability of the DNN, *i.e.*, low capacity in resisting catastrophic forgetting.

Table 2 illustrated the stability difference $\Delta FM_{\text{low}}$ between the low-order model $f_{\text{low}}$ and the baseline model $f_{\text{base}}$, as well as $\Delta FM_{\text{high}}$ between the high-order model $f_{\text{high}}$ and the baseline model $f_{\text{base}}$, *i.e.*, $\Delta FM_{\text{low}} = FM(f_{\text{low}}) - FM(f_{\text{base}})$ and $\Delta FM_{\text{high}} = FM(f_{\text{high}}) - FM(f_{\text{base}})$. We discovered that $\Delta FM_{\text{high}}$ was consistently larger than $\Delta FM_{\text{low}}$, which indicated that compared to the baseline model, the stability of the high-order model was much worse than that of the low-order model. Thus, preventing the DNN from encoding low-order interactions could significantly harm its stability in resisting catastrophic forgetting, which suggested low-order interactions might, to some extent, be an effective factor for mitigating catastrophic forgetting.

Additionally, we also observed a similar phenomenon based on different CIL models introduced in Section 3.2, *i.e.*, penalizing the learning of low-order interactions in the CIL model could significantly reduce its stability. Please refer to Appendix F for experimental results.

## 4 CONCLUSION AND DISCUSSION

In this paper, we make the first attempt to use interactions to provide a detailed explanation for catastrophic forgetting in class incremental learning, by quantifying the forgotten and preserved interactions of different complexities *w.r.t.* previous classes, explaining the effectiveness of different CIL methods in a unified view, and proposing a loss with theoretical guarantees to investigate the role of low-order interactions in resisting catastrophic forgetting.

Our work provides avenues for future exploration, such as applying our explanation to analyze other CIL problems (*e.g.*, the stability-plasticity trade-off) and other CIL settings (cf. Appendix D for a preliminary case on audio-visual CIL), which is theoretically feasible. We hope our explanation can serve as a theoretical foundation and shed new light on class incremental learning.

ETHICS AND REPRODUCIBILITY STATEMENT

This paper identifies and quantifies what is forgotten in class incremental learning, and provides a unified explanation for the effectiveness of different CIL methods, from the perspective of interactions. There are no ethical issues with this paper.

Besides, we have provided proofs for the theoretical results in Appendix C, and provided experimental details in Section 3.1, Section 3.2, Section 3.3, and Appendix H. The code will be released when the paper is accepted.

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

## A    THE USE OF LARGE LANGUAGE MODELS

Large language models (LLMs) are used exclusively for language polishing, and are not employed for information retrieval, discovery, or research ideation.

## B    RELATED WORK

**Class incremental learning.** Previous research related to class incremental learning primarily focused on proposing different effective methods to address the challenge of catastrophic forgetting, which could be further categorized into the following three types by Liang & Li (2023). Specifically, the *regularization-based methods* (Li & Hoiem, 2017; Zhao et al., 2020) mainly added a regularization constraint to the network's weight updates to prevent important network parameters of previous steps from changing too much. The *memory-based methods* (Zhou et al., 2023b; Lopez-Paz & Ranzato, 2017; Chaudhry et al., 2019) usually constructed a memory buffer to preserve knowledge of previous steps. The *expansion-based methods* (Yan et al., 2021; Wang et al., 2022; Rusu et al., 2016) dynamically expanded the network architecture for each new step to mitigate catastrophic forgetting. Despite their effectiveness, whether these methods share a common mechanism for mitigating catastrophic forgetting still remains unclear. Thus, this paper aims to use interactions to explain this shared mechanism.

However, less attention has been given to explaining catastrophic forgetting, and existing works commonly utilized accuracy-based metrics for explanation. Specifically, Chaudhry et al. (2018) calculated the difference between the highest past accuracy on a certain previous task and the accuracy on the same task evaluated by the current DNN to evaluate the forgetting of the DNN, which was further modified by Wang et al. (2024) to measure the proportion of performance the DNN forgot. Lopez-Paz & Ranzato (2017) computed the average change in accuracy on a previous task after training on a new task to measure the DNN's forgetting in the continual learning process. However, these accuracy-based metrics failed to reveal the intrinsic reason behind catastrophic forgetting, because they could not explain and identify which knowledge *w.r.t.* previous classes was forgotten and preserved in incremental learning process. Thus, they could only offer a coarse-grained explanation, leaving the fine-grained dynamics of knowledge preservation and erasure unexplored. To this end, we quantify the forgotten interactions of different complexities to provide deep insights into the core factors driving catastrophic forgetting.

Notably, the recent work (Li et al., 2025) has provided a theoretical analysis of catastrophic forgetting in a two-layer CNN, using a multi-view data model to disentangle different types of features and further tracking how these features changes during binary-classification task incremental learning. They discover that the existence of catastrophic forgetting is due to the larger signal of the task-specific feature compared to the general feature, which echoes our finding that more low-order interactions are forgotten during CIL in Section 3.1. Beyond the above similar conclusion, we further make the **first** attempt to use interactions to provide a unified understanding of different CIL methods, *i.e.,* reducing the forgetting of low-order interactions, and propose a simple-yet-efficient method to verify the primary role of low-order interactions in mitigating catastrophic forgetting.

**Interaction-based explanations of DNNs.** Although DNNs have achieved remarkable performances on different tasks, their underlying decision-making process still remains opaque and uninterpretable to humans, potentially introducing risks. Thus, explainable AI has received increasing attention in recent years, and post-hoc explanations of DNNs is a typical direction of it. However, the disappointing view of the faithfulness of post-hoc explanations of DNNs has existed for years (Adebayo et al., 2018; Rudin, 2019). Fortunately, Ren et al. (2024a); Li & Zhang (2023); Ren et al. (2023a) have proposed the interaction metric based on game theory as a new perspective to analyze the inference logic of a DNN, and the faithfulness of taking interactions as primitive inference patterns of the DNN has been mathematically ensured by a series of theorems, which also served as the the theoretical foundation of this paper.

Thus, a line of research has employed interactions to formulate and define concepts encoded by a DNN (Ren et al., 2024a; 2023a), to mathematically explain the representation capacity of DNNs (Zhou et al., 2024b; Liu et al., 2023; Ren et al., 2021; 2024b), to unify the common mechanism behind various attribution methods (Deng et al., 2024), and explain the shared mechanism behind different transferability-boosting methods (Wang et al., 2021; Zhang et al., 2022). Thus, this

paper further develops the above interaction theory system by using interactions to provide detailed explanations for catastrophic forgetting in class incremental learning, *i.e.,* quantifying the explicit forgotten knowledge, explaining the shared mechanism of classical CIL methods, and summarizing the effective factor for mitigating catastrophic forgetting.

**Comparison between interactions and the partial information decomposition.** Both interactions and partial information decomposition (PID) (Tokui & Sato, 2022; Williams & Beer, 2010; Griffith & Koch, 2014) aim to perform a disentanglement to explain the DNN, but they differ in what they decompose, how the decomposition is carried out, and what the decomposed components represent.

Specifically, PID decomposes the mutual information between the latent representation and the ground-truth generative factors into different information components based on the information theory, under the assumption that the true generative factors are available for each data point. This decomposition measures how much information of the generative factor is exclusively contained by a single latent representation and how much is shared by multiple latent representations.

In comparison, the interaction, representing the AND relationship among different input variables of an input sample, disentangle the network output into different interaction effects based on game theory, *i.e.,* $f(\boldsymbol{x}) = \sum_{S \subseteq N} I(S|\boldsymbol{x})$. This decomposition enables a detailed explanation for the inference logic of the DNN, because it attributes the output to the effects of different interactions. More crucially, the faithfulness of interactions is theoretically ensured by the sparsity property in Theorem 1 and universal-matching property in Theorem 2. Thus, we use interactions to quantify and identify which complexities of knowledge are preserved, forgotten, and newly learned during CIL, so as to provide a more fine-grained explanation for catastrophic forgetting.

**Quantifying the knowledge encoded in DNNs.** Explaining and quantifying the precise amount of knowledge in a DNN still remains a significant challenge in the field of explainable AI. A series of prior works used the mutual information between input variables and network outputs/intermediate-layer features to quantify the knowledge (Shwartz-Ziv & Tishby, 2017; Saxe et al., 2018; Higgins et al., 2017), but accurately measuring the mutual information was difficult (Kolchinsky et al., 2019).

Besides, other studies often employed human-annotated semantic concepts (Bau et al., 2017; Kim et al., 2018) or automatically learned concepts (Chen et al., 2019) to explain the knowledge encoded in the DNN, but these works lacked a mathematically guaranteed boundary that precisely defined the scope of each concept or knowledge. Thus, these studies could not quantify the exact amount of forgotten/preserved knowledge during incremental learning procedure. In comparison, the theoretically verifiable interactions allow us to represent knowledge as primitive inference patterns. In this way, we can explicitly quantify how many interactions *w.r.t.* previous steps are forgotten and preserved during class incremental learning, so as to provide detailed explanations for catastrophic forgetting.

## C    PROOF OF THEOREM 3

**Theorem 3.** *The change of the network output $\Delta f(m_1, m_2)$ is proven to be decomposed into inter-action effects of different orders.*

$$\Delta f(m_1, m_2) = \sum_{m=0}^{n} w^{(m)} \cdot \mathbb{E}_{S \subseteq N, |S|=m}[I(S|\boldsymbol{x})],$$

$$w^{(m)} = \begin{cases} C_{m_2 n}^m - C_{m_1 n}^m, & m \leq m_1 n, \\ C_{m_2 n}^m, & m_1 n < m \leq m_2 n, \\ 0, & m_2 n < m \leq n. \end{cases} \tag{12}$$

*Proof.* The difference in network outputs between different randomly masked samples is represented as:

$$\Delta f(m_1, m_2) = \mathbb{E}_{\substack{S_1, S_2 : \emptyset \subseteq S_1 \subsetneq S_2 \subseteq N \\ |S_1|=m_1 n, |S_2|=m_2 n}} [f(\boldsymbol{x}_{S_2}) - f(\boldsymbol{x}_{S_1})],$$

$$= \mathbb{E}_{\substack{S_2 \subseteq N, \\ |S_2|=m_2 n}} [f(\boldsymbol{x}_{S_2})] - \mathbb{E}_{\substack{S_1 \subseteq N, \\ |S_1|=m_1 n}} [f(\boldsymbol{x}_{S_1})] \tag{13}$$

where subsets $S_1$ and $S_2$ are randomly sampled from the universal set $N$, $0 \leq m_1 \leq m_2 < 1$.

Then, according to the Theorem 2, the first term in Eq.(2) can be re-written as follows, where $S_1 \subseteq N, |S_1| = m_1 n$.

$$\mathbb{E}[f(\boldsymbol{x}_{S_1})] = \mathbb{E}_{S_1}[\sum_{S \subseteq S_1} I(S|\boldsymbol{x})]$$

$$= \mathbb{E}_{S_1}[\sum_{m=0}^{m_1 n} \sum_{S \subseteq S_1, |S|=m} I(S|\boldsymbol{x})]$$

$$= \sum_{m=0}^{m_1 n} \mathbb{E}_{S_1}[C_{m_1 n}^m \mathbb{E}_{S \subseteq S_1, |S|=m}[I(S|\boldsymbol{x})]] \tag{14}$$

$$= \sum_{m=0}^{m_1 n} C_{m_1 n}^m \mathbb{E}_{S_1}[\mathbb{E}_{S \subseteq S_1, |S|=m}[I(S|\boldsymbol{x})]],$$

Similarly, we can obtain,

$$\mathbb{E}[f(\boldsymbol{x}_{S_2})] = \sum_{m=0}^{m_2 n} C_{m_2 n}^m \mathbb{E}_{S_2}[\mathbb{E}_{S \subseteq S_2, |S|=m}[I(S|\boldsymbol{x})]], \tag{15}$$

where $S_2 \subseteq N, |S_2| = m_2 n$. Note that $\mathbb{E}_{S_1}\mathbb{E}_{S \subseteq S_1, |S|=m}[I(S|\boldsymbol{x})]$ is averaged over all subsets $S_1 \subseteq N$ and $\mathbb{E}_{S_2}\mathbb{E}_{S \subseteq S_2, |S|=m}[I(S|\boldsymbol{x})]$ is averaged over subsets $S_2 \subseteq N$. Then, we can prove,

$$\mathbb{E}_{S_1}[\mathbb{E}_{S \subseteq S_1, |S|=m}[I(S|\boldsymbol{x})]] = \mathbb{E}_{S_2}[\mathbb{E}_{S \subseteq S_2, |S|=m}[I(S|\boldsymbol{x})]]$$

$$= \mathbb{E}_{S \subseteq N, |S|=m}[I(S|\boldsymbol{x})]. \tag{16}$$

Specifically, let us first consider $\mathbb{E}_{S_1}[\mathbb{E}_{S \subseteq S_1, |S|=m}[I(S|\boldsymbol{x})]]$, which can be re-written as follows.

$$\mathbb{E}_{S_1}[\mathbb{E}_{S \subseteq S_1, |S|=m}[I(S|\boldsymbol{x})]] = \sum_{S_1 \subseteq N, |S_1|=m_1 n} \frac{1}{\binom{n}{m_1 n}}[\frac{1}{\binom{m_1 n}{m}} \sum_{S \subseteq S_1, |S|=m} I(S|\boldsymbol{x})]$$

$$= \sum_{S \subseteq N, |S|=m} I(S|\boldsymbol{x})[\sum_{S_1 \supseteq S, |S_1|=m_1 n} \frac{1}{\binom{n}{m_1 n}} \cdot \frac{1}{\binom{m_1 n}{m}}] \quad \%\text{swaping the order of summation}$$

$$= \sum_{S \subseteq N, |S|=m} I(S|\boldsymbol{x}) \frac{\binom{n-m}{m_1 n - m}}{\binom{n}{m_1 n} \cdot \binom{m_1 n}{m}} \tag{17}$$

$$= \sum_{S \subseteq N, |S|=m} I(S|\boldsymbol{x}) \frac{\binom{n-m}{m_1 n - m}}{\binom{n}{m} \cdot \binom{n-m}{m_1 n - m}} \quad \%\text{using the combinatorial identity}$$

$$= \sum_{S \subseteq N, |S|=m} I(S|\boldsymbol{x}) \frac{1}{\binom{n}{m}}$$

$$= \mathbb{E}_{S \subseteq N, |S|=m}[I(S|\boldsymbol{x})].$$

Similarly, $\mathbb{E}_{S_2}[\mathbb{E}_{S\subseteq S_2,|S|=m}[I(S|\boldsymbol{x})]]$ can be re-written as follows.

$$
\begin{aligned}
\mathbb{E}_{S_2}[\mathbb{E}_{S\subseteq S_2,|S|=m}[I(S|\boldsymbol{x})]] &= \sum_{S_2\subseteq N,|S_2|=m_2 n} \frac{1}{\binom{n}{m_2 n}}\Big[\frac{1}{\binom{m_2 n}{m}}\sum_{S\subseteq S_2,|S|=m} I(S|\boldsymbol{x})\Big] \\
&= \sum_{S\subseteq N,|S|=m} I(S|\boldsymbol{x})\Big[\sum_{S_2\supseteq S,|S_2|=m_2 n} \frac{1}{\binom{n}{m_2 n}}\cdot\frac{1}{\binom{m_2 n}{m}}\Big] \\
&= \sum_{S\subseteq N,|S|=m} I(S|\boldsymbol{x})\frac{\binom{n-m}{m_2 n-m}}{\binom{n}{m_2 n}\cdot\binom{m_2 n}{m}} \\
&= \sum_{S\subseteq N,|S|=m} I(S|\boldsymbol{x})\frac{\binom{n-m}{m_2 n-m}}{\binom{n}{m}\cdot\binom{n-m}{m_2 n-m}} \\
&= \sum_{S\subseteq N,|S|=m} I(S|\boldsymbol{x})\frac{1}{\binom{n}{m}} \\
&= \mathbb{E}_{S\subseteq N,|S|=m}[I(S|\boldsymbol{x})].
\end{aligned}
\tag{18}
$$

Thus, we can obtain $\mathbb{E}_{S_1}[\mathbb{E}_{S\subseteq S_1,|S|=m}[I(S|\boldsymbol{x})]] = \mathbb{E}_{S_2}[\mathbb{E}_{S\subseteq S_2,|S|=m}[I(S|\boldsymbol{x})]] = \mathbb{E}_{S\subseteq N,|S|=m}[I(S|\boldsymbol{x})]$, when $m < m_1 n < m_2 n$. Thus, the output change $\Delta f(m_1, m_2)$ can be rewritten as follows:

$$
\begin{aligned}
\Delta f(m_1, m_2) &= \mathbb{E}_{S_1,S_2:\emptyset\subseteq S_1\subset S_2\subseteq N}[f(\boldsymbol{x}_{S_2}) - f(\boldsymbol{x}_{S_1})] \\
&= \mathbb{E}_{S_2\subseteq N,|S_2|=m_2 n}[f(\boldsymbol{x}_{S_2})] - \mathbb{E}_{S_1\subseteq N,|S_1|=m_1 n}[f(\boldsymbol{x}_{S_1})] \\
&= \sum_{m=0}^{m_2 n} C_{m_2 n}^m \mathbb{E}_{S_2}[\mathbb{E}_{S\subseteq S_2,|S|=m}[I(S|\boldsymbol{x})]] - \sum_{m=0}^{m_1 n} C_{m_1 n}^m \mathbb{E}_{S_1}[\mathbb{E}_{S\subseteq S_1,|S|=m}[I(S|\boldsymbol{x})]] \\
&= \sum_{m=0}^{n} w^{(m)}\mathbb{E}_{S\subseteq N,|S|=m}[I(S|\boldsymbol{x})],
\end{aligned}
\tag{19}
$$

where

$$
w^{(m)} = \begin{cases} C_{m_2 n}^m - C_{m_1 n}^m, & m \le m_1 n, \\ C_{m_2 n}^m, & m_1 n < m \le m_2 n, \\ 0, & m_2 n < m \le n. \end{cases}
\tag{20}
$$

Thus, Theorem 3 is proven. $\qquad\square$

## D    MORE RESULTS ON VERIFYING THAT THE CIL METHOD MAKES THE DNN FORGET LESS INTERACTIONS OF PREVIOUS CLASSES

In this section, we conducted experiments on another scenario of CIL, *i.e.* audio-visual CIL, to verify the generality of our explanation. Specifically, we followed settings in (Pian et al., 2023) to train two versions of autoencoders (VideoMAE for video and AudioMAE for audio) on the audio-visual dataset AVE (Tian et al., 2018) for 4 incremental steps, including a baseline model and a CIL model trained with iCaRL (Rebuffi et al., 2017). Fig. 6 shows the difference in forgotten interactions $\Delta\mathcal{I}^m_{\text{forget}}$ of different orders $m$ between the CIL model and the baseline model, which also verified our conclusion that the CIL method makes the DNN forget less interactions for previous classes to mitigate catastrophic forgetting, *i.e.*, $\mathbb{E}_m[\Delta\mathcal{I}^m_{\text{forget}}] > 0$, supporting the generality of our conclusion.

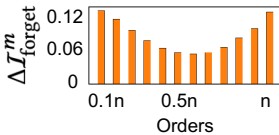

Figure 6: The difference in forgotten interactions $\Delta\mathcal{I}^m_{\text{forget}}$ *w.r.t.* previous classes between the CIL model and the baseline model, both trained on the AVE dataset for audio-visual class-incremental learning.

## E    EXPLAINING WHY MEMO MAKES THE DNN FORGET MORE HIGH-ORDER INTERACTIONS

In this section, we explain the observation that MEMO (Zhou et al., 2023b) makes its corresponding CIL model forget more high-order interactions in Fig. 4 and Fig. 3, compared to the baseline model. We consider that the CIL model trained by MEMO forgets more high-order interaction *w.r.t.* previous steps, in order to learn new interactions *w.r.t.* current classes.

To this end, we first use interactions to quantify the learning of new interactions of each complexity for the inference of current classes. Specifically, given an input sample $\boldsymbol{x}_t \in \mathcal{D}_t$ and the DNN $f_t$ incrementally learned from steps 1 to $t$, the strength of $m$-order newly learned interaction $\mathcal{I}^{(t),m}_{\text{new}}(S|\boldsymbol{x}_t)$ is defined as the interaction encoded in $f_t$ but not encoded in the previous DNN $f_{t-1}$.

$$\forall S \subseteq N, |S| = m, \quad \mathcal{I}^{(t),m}_{\text{new}}(S|\boldsymbol{x}_t) = \mathcal{I}^m(S|\boldsymbol{x}_t, f_t) - \mathcal{I}^{(t),m}_{\text{share}}(S|\boldsymbol{x}_t),$$
$$\mathcal{I}^{(t),m}_{\text{share}}(S|\boldsymbol{x}_t) = \Gamma^t_{t-1}(S|\boldsymbol{x}_t) \cdot \min(\mathcal{I}^m(S|\boldsymbol{x}_t, f_t), \mathcal{I}^m(S|\boldsymbol{x}_t, f_{t-1})),$$

(21)

where $\mathcal{I}^m(S|\boldsymbol{x}_t, f_t) \triangleq |I^m(S|\boldsymbol{x}_t, f_t)|$ denotes the strength of the $m$-order interaction $I^m(S|\boldsymbol{x}_t, f_t)$ extracted from the sample $\boldsymbol{x}_t \in \mathcal{D}_t$. $\mathcal{I}^{(t),m}_{\text{share}}(S|\boldsymbol{x}_t)$ represents the $m$-order interaction shared by $f_{t-1}$ and $f_t$. $\Gamma^t_{t-1}(S|\boldsymbol{x}_t) = \mathbb{1}((I^m(S|\boldsymbol{x}_t, f_t) \cdot I^m(S|\boldsymbol{x}_t, f_{t-1})) > 0)$ measures whether the $m$-order interaction $I^m(S|\boldsymbol{x}_t, f_t)$ encoded by the current DNN $f_t$ has the same effect as $I^m(S|\boldsymbol{x}_t, f_{t-1})$ encoded by the previous DNN $f_{t-1}$, where $\mathbb{1}(\cdot)$ is the indicator function. If $I^m(S|\boldsymbol{x}_t, f_t)$ and $I^m(S|\boldsymbol{x}_t, f_{t-1})$ have opposite effects, then the shared interaction $\mathcal{I}^{(t),m}_{\text{share}}(S|\boldsymbol{x}_t) = 0$. Otherwise, the shared interaction is quantified as $\mathcal{I}^{(t),m}_{\text{share}}(S|\boldsymbol{x}_t) = \min(\mathcal{I}^m(S|\boldsymbol{x}_t, f_t), \mathcal{I}^m(S|\boldsymbol{x}_t, f_{t-1}))$.

Then, we compare the difference in newly learned interactions $\Delta\mathcal{I}^m_{\text{learn}}$ between the baseline model and the CIL model trained by MEMO, in order to check whether the CIL model trained by MEMO learns more new high-order interactions than the baseline model.

$$\Delta\mathcal{I}^m_{\text{learn}} = \mathcal{I}^m_{\text{learn,CIL}} - \mathcal{I}^m_{\text{learn, base}}$$
$$\mathcal{I}^m_{\text{learn,base}} = \mathbb{E}^T_{t=2}\mathbb{E}_{\boldsymbol{x}_t \in \mathcal{D}_t}\mathbb{E}_{\substack{S \subseteq N, \\ |S|=m}}[\mathcal{I}^{(t),m}_{\text{new, base}}(S|\boldsymbol{x}_t)]$$

(22)

$$\mathcal{I}^m_{\text{learn,CIL}} = \mathbb{E}^T_{t=2}\mathbb{E}_{\boldsymbol{x}_t \in \mathcal{D}_t}\mathbb{E}_{\substack{S \subseteq N, \\ |S|=m}}[\mathcal{I}^{(t),m}_{\text{learn,CIL}}(S|\boldsymbol{x}_t)],$$

where $\mathcal{I}^{(t),m}_{\text{learn,CIL}}(S|\boldsymbol{x}_t)$ denotes the new interactions *w.r.t* the current step $t$ learned by the CIL model, which is obtained by applying $f_t = f_{t,\text{CIL}}$ to compute Eq. (21). Accordingly, $\mathcal{I}^{(t),m}_{\text{learn,base}}(S|\boldsymbol{x}_t)$ repre-

sents the new interactions learned by the baseline model, which is obtained by applying $f_t = f_{t,\text{base}}$ to calculate Eq. (21). Thus, a positive value of $\Delta I_{\text{learn}}^m$ indicates that the CIL model learns more $m$-order interactions *w.r.t* new classes.

Fig. 7 shows that the CIL model trained by MEMO usually learns more high-order interactions *w.r.t.* current classes, compared to the baseline model. This may partially explain MEMO makes its corresponding CIL model forget more high-order interactions *w.r.t.* previous classes.

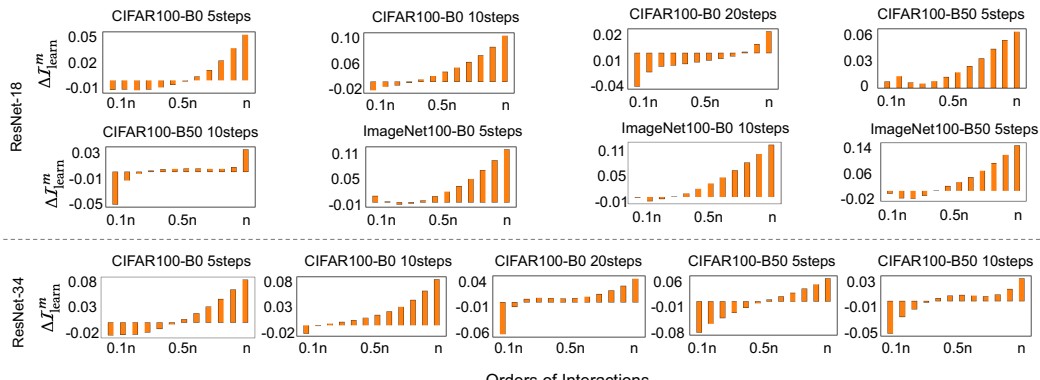

Figure 7: The difference in newly learned interactions $\Delta \mathcal{I}_{\text{learn}}^m$ between the baseline model and the CIL model trained by MEMO.

## F   MORE RESULTS ON EXPLORING THE ROLE OF LOW-ORDER INTERACTIONS

In this section, we conducted experiments to explore the role of low-order interactions in the resistance of catastrophic forgetting. To this end, we trained ResNet-18 and ResNet-34 models on CIFAR-100 datasets under different class splits in Section 3.1 for class incremental learning. For each DNN, we trained three versions, including one using a specific CIL method without any interaction penalization (*i.e.,* $\alpha = 0$, the CIL model introduced in Section 3.2), and two others with the same CIL method but applying interaction penalization ($\alpha = 1.0$), by setting $m_1 = 0, m_2 = 0.3$ and $m_1 = 0.7, m_2 = 1.0$ in $L(m_1, m_2)$, respectively. For simplicity, we named the CIL model trained to penalize $[0, 0.3n]$-order ($[0.7n, n]$-order) interactions **high-order (low-order) CIL model**, as it mainly encoded high-order (low-order) interactions. Then, we compared the stability difference $\Delta FM_{\text{CIL, low}}$ between the low-order CIL model $f_{\text{low, CIL}}$ and the CIL model $f_{\text{CIL}}$, as well as $\Delta FM_{\text{CIL, high}}$ between the high-order CIL model $f_{\text{CIL, high}}$ and the CIL model $f_{\text{CIL}}$, *i.e.,* $\Delta FM_{\text{CIL, low}} = FM(f_{\text{CIL, low}}) - FM(f_{\text{CIL}})$ and $\Delta FM_{\text{CIL, high}} = FM(f_{\text{CIL, high}}) - FM(f_{\text{CIL}})$.

Table 2 illustrated that $\Delta FM_{\text{CIL, high}}$ was consistently larger than $\Delta FM_{\text{CIL, low}}$, which indicated that the compared to the CIL, model, the stability of high-order CIL models was much worse than that of the low-order CIL model. Thus, preventing the DNN from encoding low-order interactions could significantly harm its stability in resisting catastrophic forgetting, which suggested that low-order interactions might, to some extent, be an effective factor for mitigating catastrophic forgetting.

| Model | Metric | CIFAR100-B0 5 steps | CIFAR100-B0 10 steps | CIFAR100-B0 20 steps | CIFAR100-B50 5 steps |
|---|---|---|---|---|---|
| ResNet-18 | $\Delta FM_{\text{MEMO, low}}$ | 0.08 | 0.07 | 0.08 | 0.02 |
| | $\Delta FM_{\text{Memo,high}}$ | 0.17 | 0.16 | 0.20 | 0.13 |
| ResNet-34 | $\Delta FM_{\text{MEMO,low}}$ | 0.18 | 0.1 | 0.10 | 0.10 |
| | $\Delta FM_{\text{MEMO,high}}$ | 0.30 | 0.26 | 0.23 | 0.15 |
| ResNet-18 | $\Delta FM_{\text{LWF, low}}$ | 0.20 | 0.11 | 0.06 | 0.15 |
| | $\Delta FM_{\text{LWF,high}}$ | 0.32 | 0.25 | 0.29 | 0.33 |
| ResNet-34 | $\Delta FM_{\text{LWF, low}}$ | 0.07 | 0.08 | 0.09 | 0.04 |
| | $\Delta FM_{\text{LWF, high}}$ | 0.16 | 0.21 | 0.23 | 0.18 |
| ResNet-18 | $\Delta FM_{\text{DER, low}}$ | 0.15 | 0.09 | 0.07 | 0.06 |
| | $\Delta FM_{\text{DER, high}}$ | 0.32 | 0.28 | 0.19 | 0.24 |
| ResNet-34 | $\Delta FM_{\text{DER, low}}$ | 0.07 | 0.12 | 0.15 | 0.07 |
| | $\Delta FM_{\text{DER, high}}$ | 0.18 | 0.26 | 0.42 | 0.34 |
| ResNet-18 | $\Delta FM_{\text{FOSTER, low}}$ | 0.02 | 0.05 | 0.05 | 0.09 |
| | $\Delta FM_{\text{FOSTER, high}}$ | 0.21 | 0.25 | 0.21 | 0.30 |
| ResNet-34 | $\Delta FM_{\text{FOSTER, low}}$ | 0.04 | 0.07 | 0.14 | 0.08 |
| | $\Delta FM_{\text{FOSTER, high}}$ | 0.13 | 0.29 | 0.23 | 0.26 |

Table 2: The stability difference $\Delta FM_{\text{CIL, low}}$ between the low-order CIL model and the CIL model, as well as $\Delta FM_{\text{CIL, high}}$ between the high-order CIL model and the CIL model.

## G  SPARSITY OF INTERACTIONS

Given an input sample $\boldsymbol{x}$ with $n$ input variables, among all $2^n$ possible interactions, Ren et al. (2024a) have proven that the number of interactions with salient effects on the network output is $\mathcal{O}(n^\kappa/\tau)$ under three common mathematical conditions. First, the DNN does not encode interactions of extremely high orders. Second, the average classification confidence monotonically decreases as more input variables are masked, which is computed on $\{\boldsymbol{x}_T \| |T| = n - k\}$ by masking different random sets of $k$ input variables. Third, the decreasing speed of the average confidence is polynomial. $\kappa$ is empirically within the range of $[1.9, 2.2]$ Ren et al. (2024b). This indicates that the number of salient interactions is much less than $2^n$, *i.e.,* the interactions are sparse.

To this end, we conducted experiments on incrementally trained DNNs introduced in Section 3.1 to examine whether interactions satisfy the sparsity property in real applications. Fig. 8 shows all the interaction effects encoded by the incrementally trained DNN on different input samples. We discovered that only a few interactions have salient interactions effects $|I(S|\boldsymbol{x})|$ on the network output, while the effects of all other interactions are negligible, *w.r.t.* $|I(S|\boldsymbol{x})| \approx= 0$. This finding was consistent with the conclusion found by Ren et al. (2024a), *i.e.,* interactions encoded by a DNN were usually very sparse.

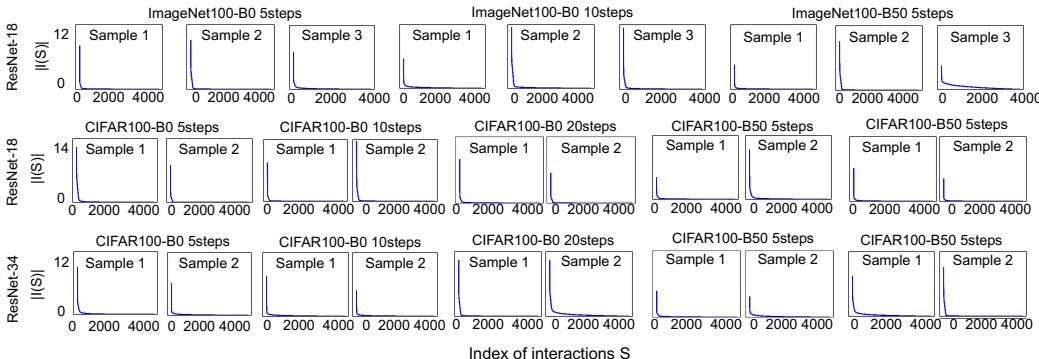

Figure 8: Interactions in descending order of the interaction strength .

## H  EXPERIMENTAL DETAILS

In this paper, we follow settings in (Li & Zhang, 2023; Ren et al., 2024a; Liu et al., 2023) to compute interactions. Considering the computational cost of calculating and extracting all $2^n$ interactions is intolerable in real implementation, we follow the method in (Li & Zhang, 2023; Ren et al., 2024a) to annotate and select a set of image patches as input variables to reduce the computation cost. Specifically, given an image in the CIFAR-100 dataset, we follow (Li & Zhang, 2023; Ren et al., 2023a) to divide it into small patches of size $4 \times 4$, resulting in a total of $8 \times 8$ image patches, and further select $n = 12$ patches from $6 \times 6$ image patches located in the center of the image as input variables to calculated interactions, because Li & Zhang (2023); Ren et al. (2024a) consider the DNN mainly used foreground information to make inference. Similarly, for each image in the ImageNet-100 dataset, we follow (Liu et al., 2023) to divide it into small patches of size $28 \times 28$, resulting in a total of $8 \times 8$ image patches, and further select $n = 12$ patches from $6 \times 6$ image patches located in the center of the image as input variables to calculated interactions. Thus, we calculate totally $2^{12} = 1024$ interactions for each input image, which costs only $2.55$ seconds per image in CIFAR-100 on ResNet-18 and $2.91$ seconds on ResNet-34, and $12.37$ seconds per image in ImageNet-100 on ResNet-18, using the single NVIDIA 4090 GPU.

Besides, to generate the masked sample $\boldsymbol{x}_T$, we follow the widely used setting in (Dabkowski & Gal, 2017; Li & Zhang, 2023; Ren et al., 2024a; Liu et al., 2023) to set the baseline value of each variable $b_i$ as the mean value of this variable across all samples in image classification to mask each input variable in $N \setminus T$. Meanwhile, given each incrementally trained DNN $f$, we follow (Li & Zhang, 2023; Ren et al., 2024a; Liu et al., 2023) to set $f(\boldsymbol{x})$ in Eq. (2) of the main paper as the confidence of predicting $\boldsymbol{x}$ to the ground-truth category $y^{\text{truth}}$, $f(\boldsymbol{x}) = \log \frac{p(y=y^{\text{truth}}|\boldsymbol{x})}{1-p(y=y^{\text{truth}}|\boldsymbol{x})}$, to com-

pute each interaction $I(S|\boldsymbol{x})$. Considering outputs of DNNs trained using different methods usually have different scales. In this way, interactions encoded in different DNNs computing based on their network outputs will have different scales, which may affect the comparison. To eliminate this impact, we follow (Cheng et al., 2024) to normalize each interaction $I(S|\boldsymbol{x})$ encoded by the DNN $f$ as $I(S|\boldsymbol{x}) \leftarrow I(S|\boldsymbol{x})/\mathbb{E}_{\boldsymbol{x}}[\|f(\boldsymbol{x}) - f(\boldsymbol{x}_\emptyset)\|]$, where $\boldsymbol{x}_\emptyset$ represents the sample with all input variables masked to baseline values. Thus, according to Theorem 2 that $f(\boldsymbol{x}) - f(\boldsymbol{x}_\emptyset) = \sum_{S \subseteq N, S \neq \emptyset} I(S|\boldsymbol{x})$, the total amount of interactions $\sum_{S \subseteq N, S \neq \emptyset} I(S|\boldsymbol{x})$ encoded by DNNs trained at different step using different methods are kept at the same magnitude, which make interactions comparable across models.

Additionally, we follow training settings in (Zhou et al., 2024a) to use SGD with an initial learning rate of 0.1 and momentum of 0.9 to train DNNs introduced in Section 2 for class incremental learning based on PyTorch and PyCIL[5] (Zhou et al., 2023a). In Section 2.4, we trained DNNs to encode interactions of specific orders based on Eq. (11), where we set $m_1 = 0, m_2 = 0.3$ and $\alpha = 1.0$ in $L(m_1, m_2)$ to train high-order DNNs, and set $m_1 = 0.7, m_2 = 1.0$ and $\alpha = 1.0$ to train low-order DNNs. This training process was roughly summarized in Algorithm 1 for a better understanding. We will release the code if the paper is accepted.

---

**Algorithm 1** Training the DNN to encode interactions of specific orders.

---

**Input**: Training dataset $\mathcal{D}_{\text{train}}$, interaction orders $m_1$ and $m_2$, coefficient $\alpha$, epoch number $E$
**Output**: a trained DNN $f$

1: Initialize parameters of $f$
2: **for** $e = 1$ to $E$ **do**
3:     Initialize $L_{\text{inter}}(m_1, m_2) = 0$ and $L_{\text{classification}} = 0$
4:     **for** $(\boldsymbol{x}, y) \in \mathcal{D}_{\text{train}}$ **do**
5:         Compute the network output change $\Delta f(m_1, m_2, \boldsymbol{x})$ based on Eq. (8)
6:         Compute the loss $L_{\text{inter}}(m_1, m_2, \boldsymbol{x}) = -\sum_{c=1}^{C}[p(\hat{y} = c|\Delta f(m_1, m_2, \boldsymbol{x})_c) \cdot \log(p(\hat{y} = c|\Delta f(m_1, m_2, \boldsymbol{x})_c))]$
7:         Compute the classification loss $L_{\text{classification}}(\boldsymbol{x}) = \text{CrossEntropy}(\boldsymbol{x}, y)$
8:         Compute $L_{\text{inter}}(m_1, m_2) += L_{\text{inter}}(m_1, m_2, \boldsymbol{x}), L_{\text{classification}} += L_{\text{classification}}(\boldsymbol{x})$
9:     **end for**
10:    Compute $L_{\text{inter}}(m_1, m_2)/ = |\mathcal{D}_{\text{train}}|, L_{\text{classification}}/ = |\mathcal{D}_{\text{train}}|$
11:    Compute the loss $L(m_1, m_2) = L_{\text{classification}} - \alpha \cdot L_{\text{inter}}(m_1, m_2)$ based on Eq. (11)
12:    Compute the gradient of the loss $L(m_1, m_2)$ to update parameters of $f$
13: **end for**

---

[5]PyCIL is an open-sourced python tool box to implement CIL.

