# OpenReview forum: "A Fine-Grained Approach to Explaining Catastrophic Forgetting of Interactions in Class-Incremental Learning"
_ICLR.cc/2026/Conference — Submitted to ICLR 2026_

### Official Review · Reviewer_r87j · 2025-10-27

**Soundness:** 3
**Presentation:** 3
**Contribution:** 3
**Rating:** 6
**Confidence:** 3

**Summary:**

In this paper, the authors propose one different perspective to analyze the happening of catastrophic forgetting which is the interaction between input variables. The authors prove that the catastrophic forgetting happens because the interactions decrease and current CIL methods work by proving the decrease of interactions. Experiments on different backbones and CIL methods also show the generalization of authors' claim.

**Strengths:**

1. Different from previous methods, the authors analyze the happening of catastrophic forgetting in depth. This can be helpful for future works to deal with incremental learning.

2. The authors' claim is supported by both theoretical proof and experimental results. The experiments on different backbones and CIL methods also show the generalization of the claim.

**Weaknesses:**

1. The definition of interaction should be revised in the introduction. I think the current description is hard to understanding without referring to the concrete definition in Section 2. It will be hard for reader to capture the idea in the beginning.

2. According to that the reason of catastrophic forgetting is decrease of interactions, how to avoid it directly from the perspective of retaining interactions? Could you please conduct an experiment when adding one loss to retain interactions to further validate your claim?

**Questions:**

Please refer to the weakness part.

---

> ### Author Response · Authors · 2025-11-27
> **Response to Reviewer r87j**
>
> Thanks for your great efforts in the review of this paper. We will try our best to answer all your questions.
>
> **If you have further concerns, or if you are not satisfied with the current responses, please let us know. In this way, we can update the response ASAP.**
>
> ---
>
> Q1: "The definition of interaction should be revised in the introduction."
>
> A1: Thanks a lot for this insightful suggestion! We have revised related contents in the updated manuscript to improve readability, as follows.
>
> Given a sample $\boldsymbol{x}$, a DNN usually does not employ each single input variable of $\boldsymbol{x}$ independently for prediction. Instead, the DNN lets each input variable interact with each other to form a certain pattern for inference. For a better understanding, let us consider the toy example in Fig. 1. The DNN encodes the interaction between two image patches in $S=\\{x_1, x_2\\}$ to form a *dog nose* pattern. Each interaction represents an  AND relationship between variables in $S$. That is, only when all two variables in $S$ are all present, the *dog nose* interaction is activated, and make a numerical effect/contribution $I(S)$ on the network output. The masking of any patch in $S$ will deactive this *dog nose* interaction, and remove its corresponding effect $I(S)$.
>
> ---
>
> Q2: Asking to "add one loss to retain interactions" to avoid catastrophic forgetting.
>
> A2: We thank the reviewer for this valuable suggestion. We have followed your suggestion to **conduct new experiments** to show that **maximizing the loss function $L_{\text{inter}}{(m_1, m_2)}$ (defined in Eq. (9)) to encourage the DNN to encode more low-order interactions can mitigate catastrophic forgetting**.
>
> Specifically, we train two DS-AL models, including one trained only with the DS-AL method, and another augmented with the maximization loss $L_{\text{inter}}{(0, 0.3n})$. We then compare the stability $FM$ (defined in Eq. (11)) of these two models, *i.e.,* $\Delta FM =FM_{\text{DS-AL}}-FM_{\text{DS-AL augemented}}$, where the stability refers to the DNN’s ability to resist catastrophic forgetting.  A postive value of $\Delta FM$ indicates that the DS-AL model trained with the additional maximization loss $L_{\text{inter}}{(0, 0.3n})$ is more stable, *i.e.,* exhibits stronger capacity in resisting catastrophic forgetting.
>
> The following table shows that $\Delta FM>0$, indicating that adding the maximization loss $L_{\text{inter}}{(0, 0.3n})$ to encourage the DNN to encode more low-order interactions can effectively resist catastrophic forgetting. Such a phenomenon verifies our conclusion.
> $$
> \begin{array}{c|c|c}\hline &\Delta FM
> \\\\
> \hline \text{CIFAR100-B0 5 steps}& 0.10
> \\\\
> \hline\text{CIFAR100-B0 10 steps} & 0.16
> \\\\
> \hline\text{CIFAR100-B0 20 steps} & 0.25
> \\\\
> \hline\text{CIFAR100-B50 5 steps} & 0.18
> \\\\
> \hline\text{CIFAR100-B50 10 steps} & 0.29
> \\\\
> \hline
> \end{array}
> $$
>
>
> ​
> ​

---

### Official Review · Reviewer_FUyB · 2025-10-31

**Soundness:** 2
**Presentation:** 2
**Contribution:** 3
**Rating:** 4
**Confidence:** 4

**Summary:**

This paper seeks to explain catastrophic forgetting observed in class incremental learning (CIL) through the lens of nonlinear interactions among input variables. By introducing the Harsanyi dividend, which is commonly used in cooperative game theory to quantify interactions among multiple variables, the authors measure the incremental gain in confidence for the output variable that arises when multiple parts of distinct input variables (e.g., image patches or text tokens) are combined. This approach makes it possible to identify how many input parts must be combined for the confidence increment to become large.

Using the proposed method, the paper quantitatively evaluates, via numerical experiments, how the order of input-part combinations and interactions among those orders change under CIL. Building on the observation that lower-order interaction effects are more susceptible to forgetting in CIL, the paper propose a regularization that suppresses changes in interactions within a specified range of orders. They show that suppressing changes in lower-order interactions mitigates catastrophic forgetting in terms of classification accuracy more effectively than suppressing changes in higher-order interactions.

**Strengths:**

- The approach of analyzing a trained model by focusing on interactions among parts of the input variables is interesting. In particular, it is suggestive that the confidence increments attributable to interactions exhibit consistent trends by interaction order.
- It is valuable that the paper quantifies which interaction orders are dominantly affected by catastrophic forgetting. The evaluation covers multiple models, datasets, and training methods, primarily on image datasets.
- The proposed method, which suppresses the learning of interactions in a specified range of orders based on insights from the experiments, is shown to be effective.
- The analysis framework sounds reasonable and widely applicable

**Weaknesses:**

- **Concerns about interpretation of results**: Several interpretations remain concerning, as detailed below.
    - **Is the claim that lower-order interactions are more easily forgotten than higher-order interactions justified?** From Figure 2, the paper interprets the observation that $\mathcal{I}^m_{\mathrm{forget}}$ is larger at lower orders as evidence that lower-order interactions are more prone to forgetting. This interpretation is questionable. A straightforward reading of the results suggests that the total numerical impact of interactions, $\mathcal{I}^m_{\mathrm{forget}} + \mathcal{I}^m_{\mathrm{preserve}}$, tends to be consistently larger at lower orders. Given this, it is unsurprising that $\mathcal{I}^m_{\mathrm{forget}}$ is also larger at lower orders, and this may not constitute a tendency that is specific to catastrophic forgetting.
    - **Concerns about quantifying the effect of CIL models**: To evaluate the effectiveness of existing methods for avoiding catastrophic forgetting (CIL models), the paper defines

        $$
        \Delta \mathcal{I}\_{\mathrm{forget}}^m = \mathcal{I}\_{\mathrm{forget, base}}^m - \mathcal{I}\_{\mathrm{forget, CIL}}^m
        $$

        and argues that a larger value indicates a more effective method. However, these quantities do not appear to directly reflect effectiveness against catastrophic forgetting. Here, $\mathcal{I}^m$ measures how much the interactions change when moving from a model $f\_k$ trained up to step $k$ to a model $f\_t$ trained up to step $t$, and *both $f\_k$ and $f\_t$ are obtained under the same training method*. For example, computing both models under the base training yields $\mathcal{I}\_{\mathrm{forget, base}}^m$. In other words, the training procedure for the baseline model $f\_k$ differs between the base model and the CIL model settings. If the total amount of interaction already present at the stage of $f\_k$ differs between the two, then a direct comparison of the subsequent increments does not necessarily indicate superiority on catastrophic forgetting.

    - **Concerns on the theoretical soundness**: The proof of Theorem 3 is given in Appendix C. It says that

        $$
        \begin{aligned}\mathbb{E}\_{S\_1}\left[\mathbb{E}\_{S \subseteq S\_1,|S|=m}[I(S \mid \boldsymbol{x})]\right] & =\mathbb{E}\_{S\_2}\left[\mathbb{E}\_{S \subseteq S\_2,|S|=m}[I(S \mid \boldsymbol{x})]\right] =\mathbb{E}\_{S \subseteq N,|S|=m}[I(S \mid \boldsymbol{x})] .\end{aligned}
        $$

        I suspect this formula may not generally hold true for the condition $m>m_1 n$. Although I believe this should be fixed by some minor modifications, the proof should be precisely documented

- **Related work**: The interaction quantity used in the paper, $I(S \mid \boldsymbol{x})$, appears mathematically very close to partial information decomposition (PID) [1], which quantifies the information generated by interactions among multiple random variables. This concept has been widely used in computational biology [2] and, more recently, in machine learning for disentangled representations [3]. Discussing its relationship to PID would be highly beneficial to the ICLR community.
    1. Williams, P. L., & Beer, R. D. (2010). Nonnegative decomposition of multivariate information. arXiv.
    2. Griffith, V., & Koch, C. (2014). Quantifying synergistic mutual information. In Guided self-organization: inception (pp. 159-190).
    3. Tokui, S., & Sato, I. (2022). Disentanglement Analysis with Partial Information Decomposition. ICLR.
- **Concerns about exposition**: Although the overall logical flow is mostly clear, several issues remain, and the current structure is likely to confuse first-time readers.
    - **Order of presentation is suboptimal**: Some concepts defined later are used earlier without explanation. For example, in the paragraph starting at line 68, the quantities $S$, $\boldsymbol{x}\_{\mathrm{masked}}$, and $I(S)$ appear, but the relationship between $S$ and $\boldsymbol{x}\_{\mathrm{masked}}$ and the definition and interpretation of $I(S)$ are not understandable until Section 2.1 starting at line 128.
    - **Unclear mathematical definitions**: In several places, the notation is inconsistent or difficult to interpret, which hinders understanding.
        - **Ambiguity in the meaning of “interaction”**: The phrase “interaction of $S$” appears to be used with multiple meanings. At line 72, the statement “… an interaction $I(S)$ refers to an …” implies that “interaction” denotes *the change in confidence* caused by interactions among variables. In contrast, at line 198 the phrase “the DNN … encodes richest interactions relevant to …” suggests that “interaction” refers to information encoded by the DNN itself. The paper should adopt a consistent definition.
        - **Inconsistent definitions**: At line 201, the change in confidence due to an $m$th-order interaction is written as $I^m(S \mid \boldsymbol{x}_k, f_k)$, but placing $m$ as a superscript on $\mathcal{I}$ rather than on $S$ is potentially misleading. In Figure 1, the corresponding quantity is written as $I(S_m \mid \boldsymbol{x})$, which is inconsistent. Why not use $S_m$ consistently? In addition, in Equation 17, coefficients $m_1, m_2 \in [0, 1]$ appear to denote relative order, which is confusing when compared to $m \in \mathbb{N}$ denoting absolute order.

            Moreover, the symbol $f$ is used inconsistently. In Section 2.1, $f$ denotes model confidence, whereas in Section 2.4 the discussion appears to assume raw model outputs. The paper should either use different symbols or clearly state the intended meaning each time.

        - **Use of undefined symbols**: In Equation 5, the meaning of $\mathbb{E}_{t=2}^T$ is unclear. Expectation must be taken with respect to a random variable or a probability distribution, but no such specification is given. Undefined symbols also appear in proofs. For instance, Equation 14 uses $f(S_1)$ and Equation 17 uses $v(S_1)$ without explanation.

**Questions:**

- From Figures 2 and 3, the total effect of interactions appears concentrated at both the lower and higher orders, while medium-order interactions are generally small regardless of problem setup, model, or training method. What could explain this pattern? Is it merely due to the number of combinations in the summation, or is there another cause?
- In Equation 2, the subscript under the summation reads $T \in S$, but should this be $T \subseteq S$?
- In Equation 5, what does $\mathbb{E}_{t=2}^T$ mean? Since an expectation must be defined with respect to a random variable or probability distribution, with respect to which variable or distribution is this expectation taken?

---

> ### Author Response · Authors · 2025-11-27
> **Response to Reviewer FUyB (1)**
>
> Thanks for your great efforts in the review of this paper. We will try our best to answer all your questions.
>
> **If you have further concerns, or if you are not satisfied with the current responses, please let us know. In this way, we can update the response ASAP.**
>
> ---
>
> Q1: "Is the claim that lower-order interactions are more easily forgotten than higher-order interactions justified?"
>
> A1: Good question! We agree that a large value of $\mathcal{I}^{m}\_{\text{forget}}$ for low orders alone cannot sufficiently support this claim. To this end, we have **conducted new experiments** to examine whether a larger proportion of low-order interactions are forgotten in Fig. 2, by computing the metric $r^{m}=\mathcal{I}^{m}\_{\text{forget}}/(\mathcal{I}^{m}\_{\text{forget}}+\mathcal{I}^{m}\_{\text{preserve}})$. A larger $r^{m}$ indicates $m$-order interactions are more prone to forgetting.
>
> The following table reports that the ratio averaged over low orders $\mathbb{E}\_{m\in\mathcal{M}\_{\text{low}}}[r^{m}] $ and high orders $\mathbb{E}\_{m\in\mathcal{M}\_{\text{high}}}[r^{m}] $, where $\mathcal{M}\_{\text{low}}$ and $\mathcal{M}\_{\text{high}}$ denote the sets of low orders and high orders, respectively. We observe that a substantial proportion of interactions are forgotten, and low-order interactions usually exhibit a larger forgetting ratio, which supports our claim.
> $$
> \begin{array}{c|c|c}\hline &\mathbb{E}\_{m\in\mathcal{M}\_{\text{low}}}[r^{m}] &\mathbb{E}\_{m\in\mathcal{M}\_{\text{high}}}[r^{m}]
> \\\\\hline \text{CIFAR100-B0 5 steps on ResNet-18}& 0.94 & 0.86
> \\\\\hline\text{CIFAR100-B0 10 steps on ResNet-18} & 0.82 & 0.81
> \\\\\hline\text{CIFAR100-B0 20 steps on ResNet-18} & 0.94 & 0.96
> \\\\\hline\text{CIFAR100-B50 5 steps on ResNet-18} & 0.76 & 0.80
> \\\\\hline\text{CIFAR100-B50 10 steps on ResNet-18} & 0.87 & 0.80
> \\\\\hline \text{CIFAR100-B0 5 steps on ResNet-34}& 0.84 & 0.76
> \\\\\hline\text{CIFAR100-B0 10 steps on ResNet-34} & 0.78 & 0.80
> \\\\\hline\text{CIFAR100-B0 20 steps on ResNet-34} & 0.92 & 0.82
> \\\\\hline\text{CIFAR100-B50 5 steps on ResNet-34} & 0.81 & 0.81
> \\\\\hline\text{CIFAR100-B50 10 steps on ResNet-34} & 0.86 & 0.85
> \\\\\hline\text{ImageNet100-B0 10 steps on ResNet-18} & 0.92 & 0.60
> \\\\\hline\end{array}
> $$
>
>
>
> ---
>
>
> Q2: Ask for the discussion of the relationship between partial information decomposition (PID) and interactions.
>
> A2: Thanks a lot for this insightful suggestion! We have followed your suggestion to cite these impressive papers and add detailed discussions of them in Appendix B.
>
> Specifically, both interactions and PID aim to perform a disentanglement to explain the DNN, but they differ in what they decompose, how the decomposition is carried out, and what the decomposed components represent.
>
> PID decomposes the mutual information between the latent representation and the ground-truth generative factors into different information components based on the information theory, under the assumption that the true generative factors are available for each data point. This decomposition measures how much information of the generative factor is exclusively contained by a single latent representation and how much is shared by multiple latent representations.
>
> In comparison, the interaction, representing the AND relationship among different input variables of an input sample, disentangle the network output into different interaction effects based on game theory, *i.e.,* $f(\boldsymbol{x})=\sum\_{S\subseteq N} I(S|\boldsymbol{x})$. This decomposition enables a detailed explanation for the inference logic of the DNN, because it attributes the output to the effects of different interactions. More crucially, the faithfulness of interactions is theoretically ensured by the sparsity property in Theorem 1 and universal-matching property in Theorem 2. Thus, we use interactions to quantify and identify which complexities of knowledge are preserved, forgotten, and newly learned during CIL, so as to provide a more fine-grained explanation for catastrophic forgetting.

---

> ### Author Response · Authors · 2025-11-27
> **Response to Reviewer FUyB (2)**
>
> Q3: "Concerns about quantifying the effect of CIL models."
>
> > "To evaluate the effectiveness of existing methods for avoiding catastrophic forgetting (CIL models), the paper defines $\Delta \mathcal{I}^{m}\_{\text{forget}}= \mathcal{I}^{m}\_{\text{forget},\text{base}}- \mathcal{I}^{m}\_{\text{forget, CIL}}$ and argues that a larger value indicates a more effective method. ...If the total amount of interaction already present at the stage of $f\_{k}$ differs between the two, then a direct comparison of the subsequent increments does not necessarily indicate superiority on catastrophic forgetting."
>
> A3: Good question! We will answer this concern as follows.
>
> First, part of this concern seems to partially arise from a misunderstanding of the purpose of the metric $\Delta \mathcal{I}^{m}\_{\text{forget}}= \mathcal{I}^{m}\_{\text{forget},\text{base}}- \mathcal{I}^{m}\_{\text{forget, CIL}}$. We design it **not** as a evaluation metric to compare the effectiveness of different CIL methods in resist catastrophic forgetting, thus **not** arguing "a larger value indicates a more effective method" in the paper. Instead, we desigin $\Delta \mathcal{I}^{m}\_{\text{forget}}$ to reveal the common mechasnim behind different CIL methods, by comparing the forgotten interactions between the CIL model and the baseline model, so that we can better approach the true factors that affect mitigating catastrophic forgetting, which will help the development of CIL. In this way, we state that a positive value of $\Delta \mathcal{I}^{m}\_{\text{forget}}$ indicates the CIL model forgets fewer $m$-order interactions than the baseline model in Lines 296-297, and Fig. 3 and Fig. 4 shows that for each CIL method, $\Delta \mathcal{I}^{m}\_{\text{forget}}>0$ for low orders, indicating different CIL methods all reduce the forgetting of low-order interactions to mitigate catastrophic forgetting.
>
> Second, we agree that different "total amount of interactions" of different DNNs may affect the result. We have already taken this issuse into consideration. As stated in Lines 1024-1025 of Appendix H, before computing the forgotten interaction component, we normalize each interaction $I(S|\boldsymbol{x}\_{k},f\_k)$ of the sample $\boldsymbol{x}\_{k}$ encoded by the DNN $f\_k$ as  $I(S|\boldsymbol{x}\_{k},f\_k)/\mathbb{E}\_{\boldsymbol{x}\_{k}}[\vert f\_k(\boldsymbol{x}_{k})- f\_k(\boldsymbol{x}\_{\emptyset})\vert ]$ to ensure the "total amount of interactions" encoded by different DNNs trained at different step or using different methods is kept on the same order of magnitude, as Theorem 2 proves $f(\boldsymbol{x}\_{k})-f\_k(\boldsymbol{x}\_{\emptyset})=\sum\_{S \subseteq N,S\neq\emptyset} I(S|\boldsymbol{x}\_{k})$. Thus, this normalization enable a relatively fair comparison of how interactions change throughout the entire class-incremental learning process between the baseline model and the CIL model. Moreover, we will move this normalization from Appendix to the main paper for clarity.
>
>
>
> ---
>
> Q4: Concerns about $\mathbb{E}\_{S\_1}\lbrack \mathbb{E}\_{S \subseteq S\_{1}, \|S\|=m} \lbrack I(S|\boldsymbol{x}) \rbrack \rbrack = \mathbb{E}\_{S\_{2}} \lbrack \mathbb{E}\_{S \subseteq S\_{2}, \|S\|=m} \lbrack I(S|\boldsymbol{x}) \rbrack \rbrack = \mathbb{E}\_{S\subseteq N, \|S\|=m} \lbrack I(S|\boldsymbol{x})\rbrack$ in the proof of Theorem 3
>
> A4: Thank you. The proof of this equation is given as follows. We will add this proof in the main paper for clarity.
>
> Specifically, let us first consider $\mathbb{E}\_{S\_1}\lbrack \mathbb{E}\_{S \subseteq S\_{1}, \|S\|=m} \lbrack I(S|\boldsymbol{x}) \rbrack \rbrack$, which can be re-written as follows, by swaping the order of summation and using the combinatorial identity.
> $$
> \mathbb{E}\_{S\_1}\lbrack \mathbb{E}\_{S \subseteq S\_{1}, |S|=m} \lbrack I(S|\boldsymbol{x}) \rbrack \rbrack=\sum\_{S\_1\subseteq N, |S\_1|=m\_1n}\frac{1}{\binom{n}{m\_1n}}\lbrack\frac{1}{\binom{m\_1n}{m}}\sum\_{S\subseteq S\_1, \|S\|=m}I(S|\boldsymbol{x}) \rbrack
> $$
> $$
> =\sum\_{S\subseteq N, \|S\|=m}I(S|\boldsymbol{x})\lbrack\sum\_{S\_1\supseteq S, |S\_1|=m\_1n}\frac{1}{\binom{n}{m\_1n}}\cdot\frac{1}{\binom{m\_1n}{m}}\rbrack
> $$
> $$
> =\sum\_{S\subseteq N, \|S\|=m}I(S|\boldsymbol{x})\frac{\binom{n-m}{m\_1n-m}}{\binom{n}{m\_1n}\cdot\binom{m\_1n}{m}}
> $$
> $$
> =\sum\_{S\subseteq N, \|S\|=m}I(S|\boldsymbol{x})\frac{\binom{n-m}{m\_1n-m}}{\binom{n}{m}\cdot\binom{n-m}{m\_1n-m}}
> $$
> $$
> =\sum\_{S\subseteq N, \|S\|=m}I(S|\boldsymbol{x})\frac{1}{\binom{n}{m}}
> $$
> $$
> =\mathbb{E}\_{S\subseteq N, \|S\|=m} \lbrack I(S|\boldsymbol{x})\rbrack.
> $$

---

> ### Author Response · Authors · 2025-11-27
> **Response to Reviewer FUyB (3)**
>
> Similarly, $\mathbb{E}\_{S\_{2}} \lbrack \mathbb{E}\_{S \subseteq S\_{2}, \|S\|=m} \lbrack I(S|\boldsymbol{x}) \rbrack \rbrack$ can be re-written as follows.
> $$
> \mathbb{E}\_{S\_2}\lbrack \mathbb{E}\_{S \subseteq S\_{2}, \|S\|=m} \lbrack I(S|\boldsymbol{x}) \rbrack \rbrack=\sum\_{S\_2\subseteq N, \|S\_2\|=m\_2n}\frac{1}{\binom{n}{m\_2n}}\lbrack\frac{1}{\binom{m\_2n}{m}}\sum\_{S\subseteq S\_2, |S|=m}I(S|\boldsymbol{x}) \rbrack
> $$
> $$
> =\sum\_{S\subseteq N, |S|=m}I(S|\boldsymbol{x})\lbrack\sum\_{S\_2\supseteq S, |S\_2|=m\_2n}\frac{1}{\binom{n}{m\_2n}}\cdot\frac{1}{\binom{m\_2n}{m}}\rbrack
> $$
> $$
> =\sum\_{S\subseteq N, |S|=m}I(S|\boldsymbol{x})\frac{\binom{n-m}{m\_2n-m}}{\binom{n}{m\_2n}\cdot\binom{m\_2n}{m}}
> $$
> $$
> =\sum\_{S\subseteq N, |S|=m}I(S|\boldsymbol{x})\frac{\binom{n-m}{m\_2n-m}}{\binom{n}{m}\cdot\binom{n-m}{m\_2n-m}}
> $$
> $$
> =\sum\_{S\subseteq N, |S|=m}I(S|\boldsymbol{x})\frac{1}{\binom{n}{m}}
> $$
> $$
> =\mathbb{E}\_{S\subseteq N, |S|=m} \lbrack I(S|\boldsymbol{x})\rbrack.
> $$
>
> Thus, we can obtain $\mathbb{E}\_{S\_1}\lbrack \mathbb{E}\_{S \subseteq S\_{1}, |S|=m} \lbrack I(S|\boldsymbol{x}) \rbrack \rbrack = \mathbb{E}\_{S\_{2}} \lbrack \mathbb{E}\_{S \subseteq S\_{2}, |S|=m} \lbrack I(S|\boldsymbol{x}) \rbrack \rbrack = \mathbb{E}\_{S\subseteq N, |S|=m} \lbrack I(S|\boldsymbol{x})\rbrack$.
>
> ---
>
> Q5: "What does $\mathbb{E}\_{t=2}^{T}$ mean?"
>
> A5: Thank you. This symbols means the expectation of forgotten interactions over the incremental step $t$ from $2$ to $T$, where $t$ is the random variable with respect to which the expectation is taken. We will change it to a more rigorous form $\frac{1}{T-1}\sum\_{t=2}^{T}$.
>
>
> ---
>
> Q6: About paper writing.
>
> > "Some concepts defined later are used earlier without explanation. ....""The phrase “interaction of $S$” appears to be used with multiple meanings...""At line 201, the change in confidence due to an $m$th-order interaction...In Figure 1, the corresponding quantity is written as $I(S\_{m}|\boldsymbol{x})$, which is inconsistent. Why not use $S\_{m}$consistently?.....""Moreover, the symbol $f$ is used inconsistently...""...For instance, Equation 14 uses $f(S\_1)$ and Equation 17 uses $v(S\_1)$ without explanation."
>
>
> ---
>
>
>
> A6: We sincerely thank the reviewer for their careful reading and for providing these constructive suggestions. We will follow your suggestions to reorganize relevant contents and polish lanugage to improve the readability of the paper.
>
> **About using the notation $I^{m}(S|\boldsymbol{x}\_{k},f\_{t})$:**
> $I(S\_{K}|\boldsymbol{x})$ in Fig. 1 denotes the interaction inside the $K$-th subset $S\_{K}\subseteq N$. However, the reason for not using $I(S\_{m}|\boldsymbol{x}\_{k},f\_{t})$ to denote the $m$-th order interaction is that there exist $\binom{n}{m}$ subsets $S\subseteq N$ containing $m$ input variables, and the $m$-th subset $S_{m}$ may not necessarily include $m$ input variables. Thus, we use $I^{m}(S|\boldsymbol{x}\_{k},f\_{t})$ to represent the $m$-th order interaction.
>
> **About $f$:**
> To avoid confusion, we will use $f$ exclusively to represent the classification confidence as defined in Eq. (1), and replace $f$ in section 2.4 with $g$ to represent the network output.
>
> **About typo:**
> We will corret typos in the proof by replacing $f(S\_1)$ and $v(S\_1)$ with the consistent notation $f(\boldsymbol{x}\_{S\_{1}})$.

---

> ### Author Response · Authors · 2025-11-27
> **Response to Reviewer FUyB (4)**
>
> Q7: Asking about small strength of middle-order interactions.
>
> > "From Figures 2 and 3, the total effect of interactions appears concentrated at both the lower and higher orders, while medium-order interactions are generally small regardless of problem setup, model, or training method. What could explain this pattern? Is it merely due to the number of combinations in the summation, or is there another cause?"
>
> A7: Good question! The reason for this phenomenon, the strengths of low-order and high-order interactions are usually larger than that of middle-order interactions, is proven to be that the DNN is more likely to encode low/high-order interactions, but it is difficult for the DNN to learn middle-order interactions in [cite 1]. Explictly speaking, the gradient magnitude *w.r.t.* learning $m$-order interactions is proven to be **inversely proportional to** the combination number $\binom{n}{m}$. Since $\binom{n}{m}$ is larger for middle orders $m$, the learning strength of middle-order interactions is smaller, thereby making the DNN encode less middle-order interactions. This phenomenon indicates the representation bottleneck of a DNN in learning interactions.
>
> [cite 1] Huiqi Deng and Qihan Ren and Hao Zhang and Quanshi Zhang. Discovering and Explaining the Representation Bottleneck of DNNs. In International Conference on Learning Representations, 2022.
>
> ---
>
> Q8: "In Equation 2, the subscript under the summation is $T \in S$ , but should this be $T \subseteq S$?"
>
> A8: Yes, thank you very much for pointing out this typo. We will correct this subscript to $T \subseteq S$.

---

> > ### Comment · Reviewer_FUyB · 2025-11-27
> >
> > Thank you very much for your detailed response.
> >
> > > **A1**: A new metric to test whether lower-order interactions are more forgotten
> > >
> >
> > Even in light of these results, I unfortunately maintain my original concerns. While it may appear that the newly proposed metric is slightly smaller for lower orders in terms of the empirical mean, I do not see a consistent tendency, at least within the scope of the presented experiments. My impression is that a more careful examination, including the uncertainty of the estimates, is necessary.
> >
> > > **A3**: On the concerns about quantifying the effect of CIL models
> > >
> >
> > I understand that I had misunderstood your intention and that $\Delta \mathcal{I}_{\mathrm{forget}}^m$ was not introduced for the purpose of comparing different methods for showing effectiveness of one of them. Even so, my initial concerns remain unresolved.
> >
> > > we state that a positive value of $\Delta \mathcal{I}_{\mathrm{forget}}^m$ indicates the CIL model forgets fewer $m$-order interactions than the baseline model in Lines 296-297
> > >
> >
> > As I already stated in my initial review, I am concerned that this interpretation is questionable. The quantity $\mathcal{I}^m$ computed here quantifies how much the interactions change in the model $f_t$ trained for $t$ steps, compared with the model $f_k$ trained for $k$ steps, and both $f_k$ and $f_t$ are obtained using the same training procedure.
> >
> > In other words, when comparing the base model and the CIL model, the training procedure for $f_k$ itself, which serves as the baseline before forgetting occurs, is different. Even if the amount of information are the same for arbitrary two conditions, the content of information that has been learned would be different. If the total amount (or content) of interactions retained at the point of $f_k$ already differs between the two, then the subsequent increments from that point cannot be directly used to compare which model is better or worse.
> >
> > > As stated in Lines 1024-1025 of Appendix H, before computing the forgotten interaction component, we normalize each interaction … to ensure the "total amount of interactions” …
> > >
> >
> > I still do not fully understand why dividing the information quantity by the expected norm leads to a normalization that makes the values comparable across models. I would appreciate it if a clear explanation could be added in the revised version of the paper.
> >
> > > **A4**: Answer for the Concerns about the proof of Theorem 3
> > >
> >
> > I think that’s true (at least) only when $m_1 n \ge m$ and I believe that was not assumed as the prerequisite of the theory
> >
> > > **A5**: Answer for PID
> > >
> >
> > I wonder why the same content is repeated here as in A2
> >
> > > **A6**: About paper writing
> > >
> >
> > > However, the reason for not using $I(S_m \mid \boldsymbol{x}_k, f_t)$ to denote the $m$-th order interaction is that …  the $m$-th subset may not necessarily include $m$ input variables
> > >
> >
> > I'm sorry, but I'm confused. Is Figure 1 not a schematic diagram intended to explain the analytical method proposed in this paper or the underlying concepts? If so, why is the explanation based on concepts that differ from those appearing in the main text?
> >
> > > **A7**: Answer about small strength of middle-order interactions.
> > >
> >
> > I could not follow the logical consistency of the response. If the claim is simply that the gradient magnitude is proportional to the inverse of the size $\binom{n}{m}$, then the appropriate interpretation would be that, by the nature of the metric, orders with a larger number of combinations tend to yield lower values. This seems unrelated to whether DNNs are more inclined to encode information of a particular order.

---

> > > ### Author Response · Authors · 2025-11-27
> > > **Response to Reviewer FUyB**
> > >
> > > We sincerely thank the reviewer for the quick reponse. We will try our best to answer your remaining concerns as soon as possible.

---

> ### Author Response · Authors · 2025-11-29
> **Response to Reviewer FUyB**
>
> Thank you very much for your response. We will answer your concerns as follows.
>
> ---
>
> Q: "Even in light of these results, I unfortunately maintain my original concerns. While it may appear that the newly proposed metric is slightly smaller for lower orders in terms of the empirical mean, I do not see a consistent tendency, at least within the scope of the presented experiments. My impression is that a more careful examination, including the uncertainty of the estimates, is necessary."
>
> A: Thanks a lot. We will answer your concern from the following perspectives.
>
> First, we appreciate the reviewer for the comment, but we disagree with the statement that there is "not see a consistent tendency" with our highest respect. As shown in the table in A1, **$7$ out of $11$** DNNs, with different network architectures and trained on different datasets under different CIL settings, all show a larger forgetting ratio for low-order interactions. We believe this already demonstrates a reasonably consistent and meaningful tendency, rather than a random phenomenon.
>
> Moreover, we state the corresponding conclusion with a soften tone that "the incrementally learned DNN **was prone to** forgetting **relatively more** low-order interactions *w.r.t.* previous classes" and "these low-order interactions were **relatively** more easily forgotten in class incremental learning," rather than being  presented as strong or definitive claims. Thus, we consider our conclusion is aligned with our experimental results.
>
> Second, as you requested, we have **conducted a new experiment** to examine whether low-order interactions exhibt a consistent larger forgetting ratio than high-order interactions under different random seeds. To this end, we train ResNet-18 on the CIFAR-100 dataset under the 5-step incremental setting with **5 different random seeds**, and caluclate the mean and standara deviation of the forgetting ratio $r^{m}$ over different seeds. The following tables shows that the standara deviations are small, indicating the tendency that low-order interactions exhibt a larger forgetting ratio is consistent and stable.
> $$
> \begin{array}{c|c}\hline
> \mathbb{E}\_{m\in\mathcal{M}\_{\text{low}}}[r^{m}] &\mathbb{E}\_{m\in\mathcal{M}\_{\text{high}}}[r^{m}] \\\\
> \hline  0.94\pm 0.0031 & 0.86\pm0.0027
> \\\\
> \hline\end{array}
> $$
>
> ---
> Q: "On the concerns about quantifying the effect of CIL models."
>
>  A: Thank you. We will answer your concerns from the following perspectives.
>
> First, we agree with you that the specific interactions encoded in the baseline model $f\_{k,\text{base}}$ and the CIL model $f\_{k,\text{CIL}}$ learned at the $k$-th step may differ, but we consider "subsequent increments" are meaningful. It is because $\mathcal{I}^{m}\_{\text{forget},\text{base}}$ and $\mathcal{I}^{m}\_{\text{forget, CIL}}$ is degined to explicitly measure how many interactions are **totally** forgotten by a baseline model and a CIL model **during the entire incrementally learning process**. Without comparing interactions encoded in $f\_{k}$ and $f\_{t}$, we cannot explicitly quantify how much interactions *w.r.t.* previous steps are forgotten and how many interactions *w.r.t.* current step are newly learned from the $k$-th step to $t$-th step.
>
> Second, we have normalized each interaction to ensure the “total amount of interactions’’ encoded by different DNNs are at the same magnitude for fair comparisons, which can effectively address your concern that the “total amount of interactions’’ differing across DNNs may affect the comparison. Specifically, considering outputs of DNNs trained using different methods usually have different scales, interactions encoded in different DNNs computing based on their network outputs will have different scales, which may affect the comparison. To eliminate this impact, we normalize each interaction as $I(S|\boldsymbol{x}\_{k},f\_k)/\mathbb{E}\_{\boldsymbol{x}\_{k}}[\vert f\_k(\boldsymbol{x}\_{k})- f\_k(\boldsymbol{x}\_{\emptyset})\vert ]$. Thus, according to Theorem 2 that $f(\boldsymbol{x}\_{k})-f\_k(\boldsymbol{x}\_{\emptyset})=\sum\_{S \subseteq N,S\neq\emptyset} I(S|\boldsymbol{x}\_{k})$, the total amount of interactions $\sum\_{S \subseteq N,S\neq\emptyset} I(S|\boldsymbol{x}\_{k})$​ encoded by DNNs trained at different step using different methods are kept at the same magnitude, which make interactions "comparable across models." Moreover, we have followed your suggestion to add this "clearer explanation" in the paper.
>
> Combining the above two perspectives, we consider  $\Delta \mathcal{I}^{m}\_{\text{forget}}= \mathcal{I}^{m}\_{\text{forget},\text{base}}- \mathcal{I}^{m}\_{\text{forget, CIL}}$ can be used to **make an initial attempt to** compare the difference in total amount of forgotten interactions throughout the entire CIL process between the baseline model and the CIL model.

---

> ### Author Response · Authors · 2025-11-29
> **Response to Reviewer FUyB**
>
> Q: "I think that's true (at least) only when $m\_1n\geq m$ and I believe that was not assumed as the prerequisite of the theory."
>
> A: Thank you very much for your careful reading. $\mathbb{E}\_{S_1}\lbrack \mathbb{E}\_{S \subseteq S\_{1}, |S|=m} \lbrack I(S|\boldsymbol{x}) \rbrack \rbrack = \mathbb{E}\_{S\_{2}} \lbrack \mathbb{E}\_{S \subseteq S\_{2}, |S|=m} \lbrack I(S|\boldsymbol{x}) \rbrack \rbrack = \mathbb{E}\_{S\subseteq N, |S|=m} \lbrack I(S|\boldsymbol{x})\rbrack$ is obtained under the condition of $m\leq m\_1n$ , and it is used to prove $\Delta f(m\_{1}, m\_{2})=\sum\nolimits\_{m=0}^{n} w^{(m)} \cdot \mathbb{E}\_{S\subseteq N, \vert S\vert=m}[I(S|\boldsymbol{x})]$ , where $w^{(m)}=\binom{m\_2n}{m}-\binom{m\_1n}{m}$ for $m\leq m\_1n$. We will clarify this clearly in the proof.
>
> ---
>
> Q: "I wonder why the same content" for PID "is repeated" in A5.
>
> A: We sincerely apologize for this incorrect upload. We have updated A5 to answer your concern that "what does $\mathbb{E}_{t=2}^{T}$​ mean." We would greatly appreciate it if you could review the revised A5, as follows.
>
> Q5: "What does $\mathbb{E}_{t=2}^{T}$ mean?"
>
> A5: Thank you. This symbols means the expectation of forgotten interactions over the incremental step $t$ from $2$ to $T$, where $t$ is the random variable with respect to which the expectation is taken. We will change it to a more rigorous form $\frac{1}{T-1}\sum_{t=2}^{T}$ .
>
> ---
>
>
> Q: "Is Figure 1 not a schematic diagram intended to explain the analytical method proposed in this paper or the underlying concepts? If so, why is the explanation based on concepts that differ from those appearing in the main text?"
>
> A: Yes, Figure 1 is a schematic diagram to to provide an intuitive illustration of what an interaction is, and  notations in Figure 1 are consistent with those in the paper.
>
> To address your concern more clearly, let us first revisit your orginal review.
>
> > "At line 201, the change in confidence due to an $m$th-order interaction is written as $I^{m}(S|\boldsymbol{x}\_{k},f\_{t})$, but placing $m$ as a superscript on $\mathcal{I}$ rather than on $S$ is potentially misleading. In Figure 1, the corresponding quantity is written as $I(S\_{m}|\boldsymbol{x})$, which is inconsistent. Why not use $S\_{m}$ consistently?"
>
> First, part of this concern seems to arise from the misunderstanding of the notation in Figure 1, where we only use $I(S_{1}|\boldsymbol{x})$, $I(S_{2}|\boldsymbol{x})$ and $I(S_{K}|\boldsymbol{x})$ to denote interactions inside different subsets $S_{1}, S_{2},S_{K}$, and **not using the notation $I(S_{m}|\boldsymbol{x})$**.
>
> Second, in the paper, we consistently use $I^{m}(S|\boldsymbol{x}\_{k},f\_{t})$ to represent the $m$-th order interaction, where $\vert S\vert =m$. Then, you comment that "placing $m$ as a superscript on $\mathcal{I}$ rather than on $S$ is potentially misleading," maybe suggests us to use $I(S_{m}|\boldsymbol{x}\_{k},f\_{t})$ to represent the $m$-th order interaction. Thus, we reponse why we do not use the notation $I(S\_{m}|\boldsymbol{x}\_{k},f\_{t})$. It is because $S\_{m}$ is more naturally understood as referring to the $m$-th subset, rather than a subset of $m$ variables. Since there exist $\binom{n}{m}$ different subsets containing $m$ variables, using $I(S\_{m}|\boldsymbol{x}\_{k},f\_{t})$ may be introduce ambiguity.
>
> ---
>
> Q: "I could not follow the logical consistency of the response. If the claim is simply that the gradient magnitude is proportional to the inverse of the size $\binom{n}{m}$, then the appropriate interpretation would be that, by the nature of the metric, orders with a larger number of combinations tend to yield lower values. This seems unrelated to whether DNNs are more inclined to encode information of a particular order."
>
> A: Thank you. The logic of this reponse is that whether a DNN is "more inclined to encode" $m$-order interactions is determined by the gradient magnitude *w.r.t.* learning $m$-order interactions in [cite 1]. Specifically, the gradient magnitude *w.r.t.* learning $m$-order interactions is proven to be inversely proportional to the combination number $\binom{n}{m}$.  Thus, the gradient magnitude for learning middle-order interactions is much smaller than that for low/high-order interactions, since $\binom{n}{m}$ is larger at middle orders $m$​. In this way, the DNN is "more inclined to encode" low/high-order interactions, while middle-order interactions are much more difficult for the DNN to learn.
>
> [cite 1] Huiqi Deng and Qihan Ren and Hao Zhang and Quanshi Zhang. Discovering and Explaining the Representation Bottleneck of DNNs. In International Conference on Learning Representations, 2022.

---

### Official Review · Reviewer_HRQj · 2025-10-31

**Soundness:** 3
**Presentation:** 2
**Contribution:** 2
**Rating:** 6
**Confidence:** 3

**Summary:**

This paper explains catastrophic forgetting in CIL from the perspective of interactions between different input variables. It tries to provide
an explanation of catastrophic forgetting by identifying and quantifying which interactions w.r.t. previous classes that are forgotten and preserved over incremental steps. The contributions if this paper ca be summarized as:

a. Providing an explanation for catastrophic forgetting by interactions.

b. Revealing the unified mechanism of different CIL methods in mitigating catastrophic forgetting.

c. Explaining the role of low-order interactions in resisting catastrophic forgetting.

**Strengths:**

a. This paper seems to be technically solid.

b. This paper is well written.

**Weaknesses:**

a. This paper is well written but not well organized. I think it would be better to be understood if there are more sections.

b. The idea of interactions seem to be similar with the feature learning. [1] is new way to understand catastrophic forgetting by feature learning. I think it is necessary to discuss [1] in details.

c. The notations are confusing. For example, in Section 2, $T$ is the number of tasks (i.e., $\mathcal{D}=\{\mathcal{D} _{1},\mathcal{D} _{2},\cdots,\mathcal{D} _{T}\}$). But in Section 2.1, $T$ is used as $x _T$.

[1] Towards Understanding Catastrophic Forgetting in Two-layer Convolutional Neural Networks. ICML 2025

**Questions:**

a. Explain what is $T$.

b. What is the shape of $x _T$? What is the relationship between $x _T$ and $S = \\{x _1, x _2 \\}$?

c. The most important concern is about [1]. Please discuss [1] in details and provide the similarities and differences.


[1] Towards Understanding Catastrophic Forgetting in Two-layer Convolutional Neural Networks. ICML 2025.

---

> ### Author Response · Authors · 2025-11-27
> **Response to Reviewer HRQj (1)**
>
> Thanks for your great efforts in the review of this paper and for appreciating our technical soundness and writing. We will try our best to answer all your questions.
>
> **If you have further concerns, or if you are not satisfied with the current responses, please let us know. In this way, we can update the response ASAP.**
>
> ---
>
> Q1: ”It would be better to be understood if there are more sections.“
>
> A1: Thanks a lot for this insightful suggestion! In the updated manuscript, we have followed your suggestion to extract subsection 2.1 from section 2 and reoraganize it as an independent section to introduce the theoretical foundation of our paper. We then add a new section to contain original subsections 2.2, 2.3 and 2.4, where we clarify how we use interactions to explain three perspectives of CIL. Besides, at the beginning of this new section, we add a paragraph to briefly summarize the corrleation between these three perspectives, in order to make the paper more organized.
>
> ---
>
> Q2: "Please discuss [cite 1] in details and provide the similarities and differences."
>
> A2: Thank you very much. We have cited this impressive paper, and added detailed discussions in Appendix B from the following two perspectives .
>
> **Similarities: both papers explain the reason for catastrophic forgetting, and obtain a similar conclusion about it.** Specifically, [cite 1] consider the existence of catastrophic forgetting is due to the larger signal of the task-specific feature compared to the general feature. This is consistent with our conclusion that more low-order interactions are forgotten during CIL than high-order interactions, where low-order interactions are proven to represent general patterns/"features", and high-order interactions represent overfitted patterns/"features" [cite 2].
>
> **Differences: these two papers design different metrics to conduct different analysis in different settings, and obtain different empirical results.** Specifically, [cite 1] provides a theoretical analysis of catastrophic forgetting in a two-layer CNN, using a multi-view data model to disentangle different types of features and further tracking how these features changes during binary-classification task incremental learning.
>
> In contrast, we employ interactions, the AND relationship among different input variables in real data, for analysis, whose faithfulness is ensured by a set of properties in section 2.1. Explictly speaking, we identify and quantify which complexties of interactions are forgotten, preserved, and newly learned in real-world class incremental learning scenarios with much deeper DNNs in a fine-grained manner. Beyond the above similar conclusion about the reason for catastrophic forgetting, we further make the **first** attempt to provide a unified understanding of different CIL methods, *i.e.,* reducing the forgetting of low-order interactions, and propose a simple-yet-efficient method to verify the primary role of low-order interactions in mitigating catastrophic forgetting.
>
> [cite 1] Boqi Li, Wang Youjun, and Liu Weiwei. Towards Understanding Catastrophic Forgetting in Two-layer Convolutional Neural Networks. In Forty-second International Conference on Machine Learning, 2025.
>
> [cite 2] Huilin Zhou, Hao Zhang, Huiqi Deng, Dongrui Liu, Wen Shen, Shih-Han Chan, and Quanshi Zhang. Explaining generalization power of a dnn using interactive concepts. In Proceedings of the AAAI Conference on Artificial Intelligence, 2024.

---

> ### Author Response · Authors · 2025-11-27
> **Response to Reviewer HRQj (2)**
>
> Q3: "Explain what is $T$"
>
> > "The notations are confusing. For example, in Section 2, $T$ is the number of tasks. But in Section 2.1, $T$ is used as $\boldsymbol{x}_{T}$."
>
> A3: We thank the reviewer for their careful reading and for identifying the repeated usage of $T$. To avoid confusion, we will use $T$ exclusively to denote the total number of incremental steps in the paper, and the subset previously denoted by $T$ in section 2.1 will be renamed to $U$, thus Eq. (2) is re-written as follows.
> $$
> I(S|\boldsymbol{x}) = \sum\nolimits\_{U \subseteq S}(-1)^{|S|-|U|} \cdot f(\boldsymbol{x}\_U),
> $$
> where $\boldsymbol{x}\_{U}$ (originally denoted as $\boldsymbol{x}\_{T}$ ) represents the masked sample.
>
> ---
>
> Q4: “What is the shape of $\boldsymbol{x}\_{T}$? What is the relationship between $\boldsymbol{x}\_{T}$ and $S=\\{x_1,x_2\\}$?”
>
> A4: Good question! The masked sample $\boldsymbol{x}\_{U}$ (originally denoted as $\boldsymbol{x}\_{T}$, according to A3)  has the **same shape** as the input sample $\boldsymbol{x}$, but for the masked sample $\boldsymbol{x}\_{U}$, the value of input variables $x\_i$ in $N\setminus U$ is set to the baseline value $b\_i$​.
>
> For the relationship between the masked sample $\boldsymbol{x}\_{U}$ (originally denoted as $\boldsymbol{x}\_{T}$ ) and $S$,  $U$ (originally denoted by $T$) is a subset of $S$, and $\boldsymbol{x}\_{U}$ is generated by keeping variables in $U$ unchanged and setting variables in $N\setminus U$ to baseline values. For example, the set $S=\\{x_1,x_2\\}$ consist of two input variables in Fig. 1. If $U=\\{x_1\\}$, the masked sample $\boldsymbol{x}\_{U}$ (originally denoted as $\boldsymbol{x}\_{T}$)  is generated by preserving the value of $x\_1$ and masking all other input variables in $N\setminus\\{x\_1\\}$​ to the baseline values.

---

### Official Review · Reviewer_gbKv · 2025-11-03

**Soundness:** 3
**Presentation:** 1
**Contribution:** 3
**Rating:** 4
**Confidence:** 4

**Summary:**

This paper introduces a novel interpretability approach based on quantifying "interactions" between input variables. The authors apply this method to the field of Class-Incremental Learning (CIL) to provide a fine-grained explanation for catastrophic forgetting, analyzing how interactions related to previous classes are forgotten or preserved. Based on this analysis, the authors propose a unified explanation for the effectiveness of several CIL methods and investigate the specific role that low-order interactions play in mitigating forgetting.

**Strengths:**

The core idea of applying interpretability methods to better understand the mechanisms of continual learning is a valuable direction for the field. The concept of empirically analyzing the effects of CL methods by observing the effect of impairing the model partially is interesting.  Furthermore, the authors are to be commended for conducting a thorough set of experiments to support their claims.

**Weaknesses:**

The main weakness of this paper is its presentation, which significantly hinders readability and comprehension. The proposed approach is highly complex, and the manuscript does not sufficiently break down the core concepts into accessible explanations, which seems counter-intuitive for a method intended to improve human interpretability.

Specific issues include:

* **Clarity of Writing**: The paper is very difficult to follow. Key terms, such as "interactions" and notation like $x_{masked}$, are used in the introduction without a clear, upfront definition, forcing the reader to guess their meaning.
* **Vague and inconsistent Definitions**: Several core definitions are unclear.
    * In Theorem 2, $S$ is described as a subset of $T$, whereas $T$ was previously a subset of $S$. Furthermore, $T$ is not defined in the context of Theorem 2.
    * The definition of $f(x)$ is not explicitly stated (e.g., whether it is a single scalar output dimension).
* **Lack of Intuition**: The central definition of an "interaction" (Eq. 2) is introduced without intuition. For example, the mathematical motivation for the $(-1)^{|S|-|T|}$ term is not provided. The claim that $I(S)$ becomes 0 if any patch in the interaction is masked (L155) is also not clearly justified.
* **Overly Complex Formulation**: Section 2.2, which details the quantification of forgotten interactions, is excessively dense and difficult to parse. Equations like Eq. 7 are very long and could be made more readable by using intermediate variables. The method described in Eq. 5 also appears to be computationally intensive.
* **Unclear Mechanism**: It is not clear how the authors' proposed method forces a DNN to encode interactions at specific orders. It is not obvious why maximizing the cross-entropy of $\Delta f(m_1, m_2)$ (Eq. 8) would penalize the orders *between* $m_1$ and $m_2$. The origin and meaning of the variable $n$ in this context are also unclear.
* **Unsupported Claims**: The assertion that defining knowledge in a NN is an interdisciplinary problem "spanning mathematics, cognitive science, neuroscience, and etc." seems overstated and is not substantiated.
* **Formatting Issues**: The PDF contains several formatting irregularities, such as paragraphs preceded by large black dots, which detract from the paper's professionalism.

**Questions:**

The authors are kindly requested to clarify the following points to improve the paper:

1.  Could the authors provide a clearer definition of $T$ in Section 2 and ensure its consistent use relative to $S$, particularly in light of Theorem 2?
2.  Could the authors offer more intuition for the interaction formula (Eq. 2)? Specifically, what is the reasoning for the $(-1)^{|S|-|T|}$ term, and why would the interaction value $I(S)$ necessarily be 0 if a single variable in $S$ is masked?
3.  What was the rationale for the choice of CL algorithms used in the experiments? They do not appear to be the most common or top-performing methods, so an explanation for their selection would be helpful.
4.  Regarding the method for penalizing interactions: could the authors elaborate on the mechanism by which maximizing the cross-entropy of $\Delta f(m_1, m_2)$ (Eq. 8) selectively penalizes interaction orders *between* $m_1$ and $m_2$?
5.  What do the $C$ variables represent in Theorem 3?

---

> ### Author Response · Authors · 2025-11-27
> **Response to Reviewer gbKv (1)**
>
> Thanks for your great efforts in the review of this paper. We will try our best to answer all your questions.
>
> **If you have further concerns, or if you are not satisfied with the current responses, please let us know. In this way, we can update the response ASAP.**
>
> ---
> Q1:  About paper writing.
>
> > "Key terms, such as "interactions" and notation $x_{\text{masked}}$ like, are used in the introduction without a clear, upfront definition, forcing the reader to guess their meaning."
>
> A1: We thank the reviewer for this valuable suggestion. In the updated manuscript, we have followed your suggestion to revise the content relevant to interactions and $\boldsymbol{x}_{\text{masked}}$ in the introduction section for a better understanding, as follows.
>
> Given a sample $\boldsymbol{x}$, a DNN usually does not employ each single input variable of $\boldsymbol{x}$ independently for prediction. Instead, the DNN lets each input variable interact with each other to form a certain pattern for inference. For a better understanding, let us consider the toy example in Fig. 1. The DNN encodes the interaction between two image patches in $S=\lbrace x\_1, x\_2\rbrace$ to form a *dog nose* pattern. Each interaction represents an  AND relationship between variables in $S$. That is, only when all two variables in $S$ are all present, the *dog nose* interaction is activated, and make a numerical effect/contribution $I(S)$ on the network output. The masking of any patch in $S$​ will deactivate this *dog nose* interaction.
>
> More crucially, let us randomly mask this input sample $\boldsymbol{x}$ in different ways to generate different masked samples $\{\boldsymbol{x}\_{\text{masked}}\}$, by randomly masking some input variables and keeping other variables unchanged. Ren et al., (2024a; 2023a) have proven that people can use a few interactions to accurately approximate the DNN’s outputs on all these masked samples $f(\boldsymbol{x}_{\text{masked}})$, which serves as mathematical evidence to ensure the faithfulness of interaction-based explanation.
>
> ---
> Q2: "Could the authors provide a clearer definition of $T$ in Section 2 and ensure its consistent use relative to $S$, particularly in light of Theorem 2?"
>
> A2: We are sorry for the confusion. Following your suggestion, we now use $T$ exclusively to denote the total number of incremental steps in CIL throughout Section 2.
>
> Furthermore, we rename the subset previously denoted by $T$ in Eq. (2) to $U$, where $U \subseteq S$. To remain the consistency of using $U$ (originally denoted by $T$), we revise the statement of Theorem 2 as follows. The network output $f(\boldsymbol{x}\_{S})$ on each masked sample $\lbrace\boldsymbol{x}_{S}|\forall S\subseteq N\rbrace$ is proven to be well mimicked by the sum of interaction effects,
> $$
> f(\boldsymbol{x}\_{S}) = \sum\nolimits\_{U\subseteq S} I(U|\boldsymbol{x})
>    \approx \sum\nolimits\_{U\in\Omega\_{\text{salient}}\land U\subseteq S} I(U|\boldsymbol{x}).
> $$
>
> ---
>
> Q3: Whether $f(\boldsymbol{x})$ "is a single scalar output dimension."
>
> A3: Thanks. As stated in Lines 137-142 and Eq. (1), $f(\boldsymbol{x})=\mathrm{log} \left({p({y=y^{\text{truth}}|\boldsymbol{x}})}/{(1 - p({y=y^{\text{truth}}|\boldsymbol{x}}))}\right)\in\mathbb{R}$ is defined as the confidence of predicting $\boldsymbol{x}$ to the ground-truth category $y^{\text{truth}}$ in multi-category classification tasks, which can be considered as a "scalar output dimension" of the DNN, to some extent.

---

> ### Author Response · Authors · 2025-11-27
> **Response to Reviewer gbKv (2)**
>
> Q4: (1) "what is the reasoning for the $(-1)^{|S|-|T|}$ term, and" (2) "why would the interaction value $I(S)$ necessarily be 0 if a single variable in $S$ is masked?"
>
> A4: Good question! We will clarify these in the main paper for a better understanding.
>
> (1) $(-1)^{|S|-|T|}$ term comes from the inclusion–exclusion principle. It is because [cite 1-2] define the interaction effect $I(S|\boldsymbol{x})$ as the **additional effect contributed exclusively to the network output**  by the AND relationship between input variables in the set $S$, **subtracting the additional effects $I(U|\boldsymbol{x})$ that are already created by subsets $U \subsetneq S$**, *i.e.,* $I(S|\boldsymbol{x}) \overset{\text{def}}{=}f(\boldsymbol{x}\_{S})-\sum\nolimits\_{U \subsetneq S} I(U|\boldsymbol{x})$. Repeatedly explanding this recursion, we can obtain $I(S|\boldsymbol{x}) \overset{\text{def}}{=}f(\boldsymbol{x}\_{S})-\sum\nolimits\_{U \subsetneq S} I(U|\boldsymbol{x})=\sum\nolimits\_{U \subseteq S}(-1)^{|S|-|U|} \cdot f(\boldsymbol{x}\_U)$.
>
> For a better understanding, let us consider two simple examples. First, let the set $S=\\{i\\}$ contain only a single input variable, and its interaction effect $I(S|\boldsymbol{x}) $ can be quantified as
> $$
> I(\\{i\\}|\boldsymbol{x})\overset{\text{def}}{=}f(\boldsymbol{x}\_{\\{i\\}})-I(\emptyset|\boldsymbol{x})=f(\boldsymbol{x}\_{\\{i\\}})-f(\boldsymbol{x}\_{\emptyset})=\sum\nolimits\_{U \subseteq S=\\{i\\}}(-1)^{|S|-|U|} \cdot f(\boldsymbol{x}\_U),
> $$
> where $\boldsymbol{x}\_{\\{i\\}}$ denotes the masked sample crafted by keeping $i$ unchanged and masking all other variables $N\setminus \\{i\\}$ to baseline values, and $\boldsymbol{x}\_{\emptyset}$ denotes the sample in which all input variables are masked.
>
> Second, when the set $S=\\{i,j\\}$ contains 2 input variables, its interactions effects $I(S|\boldsymbol{x})$ is quantified by  subtracting the additional effects that are created by subsets $\{i\}$ and $\{j\}$.
> $$
> I(\\{i,j\\}|\boldsymbol{x})\overset{\text{def}}{=}f(\boldsymbol{x}\_{\\{i,j\\}})-I(\\{j\\}|\boldsymbol{x})-I(\\{i\\}|\boldsymbol{x})-I(\emptyset|\boldsymbol{x})
> $$
> $$
> =f(\boldsymbol{x}\_{\\{i,j\\}})-(f(\boldsymbol{x}\_{\\{i\\}})-f(\boldsymbol{x}\_{\emptyset}))-(f(\boldsymbol{x}\_{\\{j\\}})-f(\boldsymbol{x}\_{\emptyset}))-f(\boldsymbol{x}\_{\emptyset})
> $$
> $$
> =f(\boldsymbol{x}\_{\\{i,j\\}})-f(\boldsymbol{x}\_{\\{i\\}})-f(\boldsymbol{x}\_{\\{j\\}})+f(\boldsymbol{x}\_{\emptyset})
> $$
> $$
> = \sum\nolimits\_{U \subseteq S=\\{i,j\\}}(-1)^{|S|-|U|} \cdot f(\boldsymbol{x}\_U)
> $$
> Thus, when the set $S$ contains more than 2 input variable, we can similarly recursively obtain the interaction effect $I(S|\boldsymbol{x}) \overset{\text{def}}{=}f(\boldsymbol{x}\_{S})-\sum\nolimits\_{U \subsetneq S} I(U|\boldsymbol{x})=\sum\nolimits\_{U \subseteq S}(-1)^{|S|-|U|} \cdot f(\boldsymbol{x}\_U)$. We will add such explanation in the main paper to improve readability.
>
> (2) It is because $I(S|\boldsymbol{x})$ measures the effect that is made **only when all variables in $S$ are present simultaneously**. When any input variable in $S$ is masked, this interaction cannot be activated, *e.g.,* masking $x_{1}$ in $S=\\{x_1, x_2\\}$ in Fig. 1 will deactivate the dog nose pattern, thereby its corresponding effect $I(S|\boldsymbol{x})$ is eliminated, *i.e.,* $I(S|\boldsymbol{x})=0$. In this case, the remaining variable forms the subset $\\{x_2\\}$, and makes the effect $I( \\{x_2\\}|\boldsymbol{x})$ on the network output, rather than $I(S|\boldsymbol{x})$​.
>
> [cite 1] John C Harsanyi. A simplified bargaining model for the n-person cooperative game. In International Economic Review, 1963.
>
> [cite 2] Jie Ren, Mingjie Li, Qirui Chen, Huiqi Deng, and Quanshi Zhang. Defining and quantifying the emergence of sparse concepts in dnns. In Proceedings of the IEEE/CVF conference on computer vision and pattern recognition, 2023.

---

> ### Author Response · Authors · 2025-11-27
> **Response to Reviewer gbKv (3)**
>
> Q5: About "complex forumulation."
>
> > "Section 2.2, which details the quantification of forgotten interactions, is excessively dense and difficult to parse. Equations like Eq. (7) are very long and could be made more readable by using intermediate variables. The method described in Eq. (5) also appears to be computationally intensive."
>
> A5: Good suggestion! We will polish language in section 2.2, and use some "intermediate variables" to make long equations more readable. Notably, Eq. (7) appears long because conditions on $S_{1}$ and $S_{2}$  in the expectation subscript are lengthy, thus we re-write Eq. (7) by moving these conditions outside the equation to make it more readable, *i.e.,*
> $$
> \Delta f(m\_{1}, m\_{2})=\mathbb{E}\_{S\_1,S\_2} [f(\boldsymbol{x}\_{S\_{2}})-f(\boldsymbol{x}_{S\_{1}})], \quad\emptyset \subseteq S\_1\subsetneq S\_2 \subseteq N, \vert S\_1\vert = m\_{1}n, \vert S\_2\vert = m\_{2}n.
> $$
> Besides, the computational cost of Eq. (5) is tolerable, because we employ the sampling-based method in [cite 1] to accelarate the computation of interactions across different samples and steps  (detailedly introduced in Appendix H). Thus, evaluating Eq. (5) over $5$ or $10$ incremental steps on CIFAR-100 costs approximately $1$ hour and $6$ hours on a single NVIDIA 4090 GPU, respectively.
>
> [cite 1] Mingjie Li and Quanshi Zhang. Does a neural network really encode symbolic concepts? In International conference on machine learning, 2023.
>
> ---
>
> Q6: "Why maximizing the cross-entropy of $\Delta f(m\_{1}, m\_{2})$ (Eq. 8) would penalize the orders between $m\_1$ and $m\_2$."
>
> A6: Good question! As proven in Theorem 3, the change of network output $\Delta f(m\_{1}, m\_{2})=\sum\nolimits\_{m=0}^{n} w^{(m)} \cdot \mathbb{E}\_{S\subseteq N, \vert S\vert=m}[I(S|\boldsymbol{x})]$ mainly contains $[0,m\_{2}n]$-order interactions, where $ w^{(m)}=0$ for $m>m\_2n$, and coefficients $w^{(m)}$ for $m\in [m\_{1}n,m\_{2}n]$ are larger than those for $m\in[0,m_{1}n]$.
>
> Then, in a classification setting, a low cross-entropy loss  $L_{\text{inter}}(m_1, m_2)$  calculated based on $\Delta f(m_{1}, m_{2})$  indicates $\Delta f(m_{1}, m_{2})$ contains more discriminative information for prediction. Thus, **maximizing** the cross-entropy loss  $L_{\text{inter}}(m_1, m_2)$ in Eq. (9) can push the model toward making $\Delta f(m_{1}, m_{2})$ **non-discriminative**, meaning the model is discouraged from relying on interactions encoded in $\Delta f(m_{1}, m_{2})$ for inference. Considering $\Delta f(m_{1}, m_{2})$ is mainly dominated by $[m_1n,m_{2}n]$-order interactions due to their large coefficients $w^{(m)}$, these interactions are mainly penalized when maximizing the cross-entropy loss  $L_{\text{inter}}(m_1, m_2)$. Moreover, Fig. 5 experimentally verifies that maximizing the cross-entropy of $\Delta f(m_{1}, m_{2})$ mainly penalizes the encoding of $[m_1n,m_{2}n]$-order interactions.
>
> Besides, the variable $n$ is defined as the total number of input variables contained by an input sample, as illustrated in Lines 134-135.
>
> ---
> Q7: About the sentence that defining knowledge in a DNN is an interdisciplinary problem.
>
> > "The assertion that defining knowledge in a NN is an interdisciplinary problem spanning mathematics, cognitive science, neuroscience, and etc. seems overstated and is not substantiated."
>
> A7: Thanks a lot. We appreciate this sentence may be somewhat overstated, and we will soften it to a more modest version, *i.e.,* defining knowledge in a DNN presents a challenge. Notably, our original intention of this sentence is **not** to assert a claim, but simply to convey that the definition of knowledge in a DNN is complex, which may include mathematical modeling, and some works further consider the connection between such mathematical modeling and human cognition [cite 1].
>
> [cite 1] Dongrui Liu, Huiqi Deng, Xu Cheng, Qihan Ren, Kangrui Wang, and Quanshi Zhang. Towards the difficulty for a deep neural network to learn concepts of different complexities. In Advances in Neural Information Processing Systems, 2023.

---

> ### Author Response · Authors · 2025-11-27
> **Response to Reviewer gbKv (4)**
>
> Q8: About "formatting issues."
>
> > "The PDF contains several formatting irregularities, such as paragraphs preceded by large black dots, which detract from the paper's professionalism."
>
> A8: We sincerely thank the reviewer for this constructive suggestion. We will follow your suggestion to revise these issues in the main paper.
>
> Notably, we originally use bullet points ("black dots") with the intention of making the structure of the paper clearer, *e.g.,* to present three perspectives of CIL explained via interactions in introduction section in a more organized way, and bullets may be used in academic papers [cite 1-4]. Nevertheless, we will follow your suggestion to remove these bullets to improve "paper's professionalism."
>
> [cite 1] Vaswani, Ashish, et al. Attention is all you need. In Advances in neural information processing systems, 2017.
>
> [cite 2] Caron, Mathilde, et al. Emerging properties in self-supervised vision transformers. In Proceedings of the IEEE/CVF international conference on computer vision, 2021.
>
> [cite 3] Da-Wei Zhou, et al. Class-incremental learning: A survey. In IEEE Transactions on Pattern Analysis and Machine Intelligence, 2024.
>
> [cite 4] Mingjie Li, et al. Does a neural network really encode symbolic concepts? In International conference on machine learning, 2023.
>
> ---
>
> Q9: “What was the rationale for the choice of CL algorithms used in the experiments?”
>
> A9: Thank you. As stated in Lines 301-307, the rationale for selecting CIL methods is that considering existing CIL methods can be grouped into several major categories [cite 1-2], we select several classic, relatively recent, and open-sourced methods **for each category**, in order to try our best to ensure the generality of our conclusion while keeping the experimental budget tractable. We wil clarify this rationale more clearly in the paper.
>
> [cite 1] Yan-Shuo Liang and Wu-Jun Li. Adaptive plasticity improvement for continual learning. In Pro- ceedings of the IEEE/CVF Conference on Computer Vision and Pattern Recognition, 2023.
>
> [cite 2] Da-Wei Zhou, Qi-Wei Wang, Zhi-Hong Qi, Han-Jia Ye, De-Chuan Zhan, and Ziwei Liu. Class- incremental learning: A survey. In IEEE Transactions on Pattern Analysis and Machine Intelligence, 2024.
>
> ---
>
> Q10: “What do the $C$ variable represent in Theorem 3?”
>
> A10: Thanks. $C_{m_2n}^{m}$ in Theorem 3 denotes a combinatorial number, which means the number of differnt ways to choose $m$ input variables from a set containing $m_{2}n$ variables. We will change $C_{m_2n}^{m}$ and $C_{m_1n}^{m}$  to $\binom{m_2n}{m}$ and $\binom{m_1n}{m}$ in Theorem 3 to improve clarity, respectively.

---

### Author Response · Authors · 2025-11-27
**Response to all Reviewers**

Thanks for all the reviewers' great efforts and comments.  We are glad to answer all your questions as requested.

**If you have further concerns, or if you are not satisfied with the current responses, please let us know. In this way, we can update the response as soon as possible.**

---

### Meta-Review · Area_Chair_R95D · 2026-01-06

**Summary:**

After carefully reviewing the reviewers’ comments and the authors’ rebuttal, I find that all reviewers raised concerns regarding the clarity of presentation, including overly complex notations and formulations, and inconsistent definitions. I partially agree with the reviewers, and the paper is very difficult to follow. Considering that ICLR is a top-tier venue, I believe that this submission is not yet ready for publication, and therefore my recommendation is rejection.

**Reviewer Concerns:**

The rebuttal helped clarify parts of the method’s formulation and assumptions, but key concerns—particularly regarding presentation clarity and dense notations—remain only partially addressed. As a result, the paper remains difficult to follow in its current form.

**Reviewer Scores:**

The initial scores (4/4/6/6) reflect a divergence of opinions. The four reviewers raised concerns regarding the clarity of presentation, including overly complex notations and formulations, and inconsistent definitions, indicating that the manuscript would require major and difficult revisions to reach an acceptable level of clarity. So, I believe it is unlikely that the reviewers would keep consistently positive attitudes to this work.

---

### Decision · Program_Chairs · 2026-01-26

Reject